# The ecology and epidemiology of malaria parasitism in wild chimpanzee reservoirs

Erik J. Scully[1,2], Weimin Liu[3], Yingying Li[3], Jean-Bosco N. Ndjango[4], Martine Peeters[5], Shadrack Kamenya[6], Anne E. Pusey [7], Elizabeth V. Lonsdorf[8], Crickette M. Sanz [9,10], David B. Morgan[11], Alex K. Piel[12], Fiona A. Stewart [12,13], Mary K. Gonder [14], Nicole Simmons[15], Caroline Asiimwe[16], Klaus Zuberbühler[17,18], Kathelijne Koops [19], Colin A. Chapman[20,21], Rebecca Chancellor[22,23], Aaron Rundus[23], Michael A. Huffman[24], Nathan D. Wolfe[25], Manoj T. Duraisingh [2✉], Beatrice H. Hahn [3✉] & Richard W. Wrangham[1✉]

Chimpanzees (*Pan troglodytes*) harbor rich assemblages of malaria parasites, including three species closely related to *P. falciparum* (sub-genus *Laverania*), the most malignant human malaria parasite. Here, we characterize the ecology and epidemiology of malaria infection in wild chimpanzee reservoirs. We used molecular assays to screen chimpanzee fecal samples, collected longitudinally and cross-sectionally from wild populations, for malaria parasite mitochondrial DNA. We found that chimpanzee malaria parasitism has an early age of onset and varies seasonally in prevalence. A subset of samples revealed *Hepatocystis* mitochondrial DNA, with phylogenetic analyses suggesting that *Hepatocystis* appears to cross species barriers more easily than *Laverania*. Longitudinal and cross-sectional sampling independently support the hypothesis that mean ambient temperature drives spatiotemporal variation in chimpanzee *Laverania* infection. Infection probability peaked at ~24.5 °C, consistent with the empirical transmission optimum of *P. falciparum* in humans. Forest cover was also positively correlated with spatial variation in *Laverania* prevalence, consistent with the observation that forest-dwelling Anophelines are the primary vectors. Extrapolating these relationships across equatorial Africa, we map spatiotemporal variation in the suitability of chimpanzee habitat for *Laverania* transmission, offering a hypothetical baseline indicator of human exposure risk.

[1] Department of Human Evolutionary Biology, Harvard University, Cambridge, MA 02138, USA. [2] Department of Immunology & Infectious Diseases, Harvard T. H. Chan School of Public Health, Boston, MA 02115, USA. [3] Department of Medicine, Perelman School of Medicine, University of Pennsylvania, Philadelphia, PA 19104, USA. [4] Department of Ecology and Management of Plant and Animal Resources, Faculty of Sciences, University of Kisangani, BP 2012 Kisangani, Democratic Republic of the Congo. [5] Recherche Translationnelle Appliquée au VIH et aux Maladies Infectieuses, Institut de Recherche pour le Développement, University of Montpellier, INSERM, 34090 Montpellier, France. [6] Gombe Stream Research Centre, The Jane Goodall Institute, Tanzania, Kigoma, Tanzania. [7] Department of Evolutionary Anthropology, Duke University, Durham, NC 27708, USA. [8] Department of Psychology, Franklin and Marshall College, Lancaster, PA 17604, USA. [9] Department of Anthropology, Washington University in St. Louis, St Louis, MO 63130, USA. [10] Congo Program, Wildlife Conservation Society, BP 14537 Brazzaville, Republic of the Congo. [11] Lester E. Fisher Center for the Study and Conservation of Apes, Lincoln Park Zoo, Chicago, IL 60614, USA. [12] Department of Anthropology, University College London, 14 Taviton St, Bloomsbury, WC1H OBW London, UK. [13] School of Biological and Environmental Sciences, Liverpool John Moores University, Liverpool L3 3AF, UK. [14] Department of Biology, Drexel University, Philadelphia, PA 19104, USA. [15] Zoology Department, Makerere University, P.O. Box 7062 Kampala, Uganda. [16] Budongo Conservation Field Station, Masindi, Uganda. [17] School of Psychology and Neuroscience, University of St Andrews, St Andrews, UK. [18] Department of Comparative Cognition, Institute of Biology, University of Neuchâtel, Neuchâtel, Switzerland. [19] Department of Ape Behaviour & Ecology Group, University of Zurich, Zurich, Switzerland. [20] Department of Anthropology, Center for the Advanced Study of Human Paleobiology, George Washington University, Washington, DC, USA. [21] School of Life Sciences, University of KwaZulu-Natal, Scottsville, Pietermaritzburg, South Africa. [22] Department of Anthropology & Sociology, West Chester University, West Chester, PA, USA. [23] Department of Psychology, West Chester University, West Chester, PA, USA. [24] Center for International Collaboration and Advanced Studies in Primatology, Primate Research Institute, Kyoto University, Inuyama, Aichi, Japan. [25] Metabiota Inc, San Francisco, CA, USA. ✉email: mduraisi@hsph.harvard.edu; bhahn@pennmedicine.upenn.edu; wranghamrichard@gmail.com

African great apes harbor a wide diversity of malaria parasites, including seven species closely related to *Plasmodium falciparum* (sub-genus *Laverania*), the most prevalent and malignant malaria parasite of humans[1–6]. Although the emergence of *P. falciparum* in the human population has been traced to a gorilla parasite[6–8], *Laverania* species appear to exhibit considerable host-specificity in wild ape populations[9–11]. Chimpanzees (*Pan troglodytes*)—the most abundant and widely distributed great ape species[12]—host three *Laverania* parasites (*P. reichenowi*, *P. gaboni*, and *P. billcollinsi*)[2,5,7,9,13–15], which are primarily transmitted by forest-dwelling *Anopheles* mosquitoes (*An. marshallii*, *An. moucheti*, *An. vinckei*) that feed promiscuously upon both humans and wild apes[16,17]. Although habitats of humans and chimpanzees often interface and humans are therefore likely to be exposed to these zoonotic parasites, three surveys of rural human populations in West Central Africa have failed to document cross-species transmission[10,11,18], suggesting that strong molecular (e.g., binding affinity of parasite invasion ligand Rh5 to host receptor basigin) as well as other barriers govern the host-specificity of the *Laverania*[19–22]. Cross-species transmission of *Laverania* has, however, recently been documented among apes living in captive environments[23], suggesting that a high magnitude of exposure can overcome molecular barriers to infection.

The capacity to identify potential exposure hotspots is currently limited because the ecological conditions that mediate *Plasmodium* transmission among chimpanzee hosts remain largely unexplored. Previous studies employing non-invasive sampling and molecular diagnostics have shown that the prevalence of chimpanzee *Laverania* infection varies regionally within equatorial Africa (generally ~30–50% among chimpanzee subspecies, though some sites exhibit an absence of infection)[7,24]. Infections are also temporally dynamic, manifesting seasonal and inter-annual trends in prevalence[25]. However, to our knowledge, no study has quantified the ecological drivers of this variation in infection probability. Doing so would offer a mechanistic explanation of these spatiotemporal dynamics and could pave the way to explicitly quantify the magnitude of human exposure in ecologically permissive settings.

Although the ecological drivers of chimpanzee *Laverania* infection remain largely unknown, both laboratory and field studies have revealed that ambient temperature is a strong determinant of *Plasmodium* transmission dynamics in human populations[26–29]. Ambient temperature influences both the population dynamics of the *Anopheles* mosquito vector and the rate of parasite development therein (i.e., the duration of sporogony)[26,30,31]. While higher temperatures accelerate the rate of parasite development within the vector, mosquito survival tends to decline above a particular temperature threshold[26,32]. Taken together, empirical studies indicate that transmission of *P. falciparum* peaks at a mean temperature of ~25 °C in human populations[27,33]. In essence, this optimal temperature represents the value at which the maximal number of mosquito vectors survive long enough to harbor the development of infectious parasites, and deviation from this value is associated with a reduction in the magnitude of transmission. Thus, spatial variation in ambient temperature defines the suitability of a particular locale for transmission, and temporal variation in this parameter underlies the seasonality of infection[27,34–36].

Additional ecological variables can influence transmission dynamics. Intra-day fluctuations in ambient temperature can amplify or dampen transmission relative to that which would be expected at a particular mean temperature[28,37]. The effect of precipitation on malaria transmission is more complex. Rainfall is necessary to support breeding habitat for aquatic mosquito larvae (e.g., transitory oviposition sites)[26,29], whereas high levels of precipitation may flush breeding sites, resulting in larval mortality[38]. Finally, given that chimpanzee malaria parasites are transmitted by forest-dwelling Anophelines[16,17], the availability of forested habitat is also likely to be a necessary determinant of infection[39], though the relationship between deforestation and human malaria parasitism has proven to be complex[40].

In this study, we characterize the ecology and epidemiology of malaria infection in wild chimpanzee reservoirs and test the hypothesis that ambient temperature constitutes a critical driver of spatiotemporal variation in chimpanzee *Laverania* prevalence. We used molecular diagnostic assays to screen chimpanzee fecal specimens, collected via both longitudinal and cross-sectional sampling strategies (Fig. 1), for malaria parasite mitochondrial DNA. These analyses revealed that the epidemiology of chimpanzee malaria parasitism is characterized by early age of onset and seasonality of infection. In addition to *Plasmodium* parasites, we amplified *Hepatocystis* (a malaria parasite of monkeys and bats[41]) from a subset of samples. While *Laverania* parasites exhibit considerable host-specificity, phylogenetic analyses revealed that *Hepatocystis* appears to cross species barriers more easily. We also analyzed the relationship between environmental variables and fecal parasite rates to define the ecological niche of the chimpanzee *Laverania*. Longitudinal and cross-sectional sampling strategies each independently supported the hypothesis that mean ambient

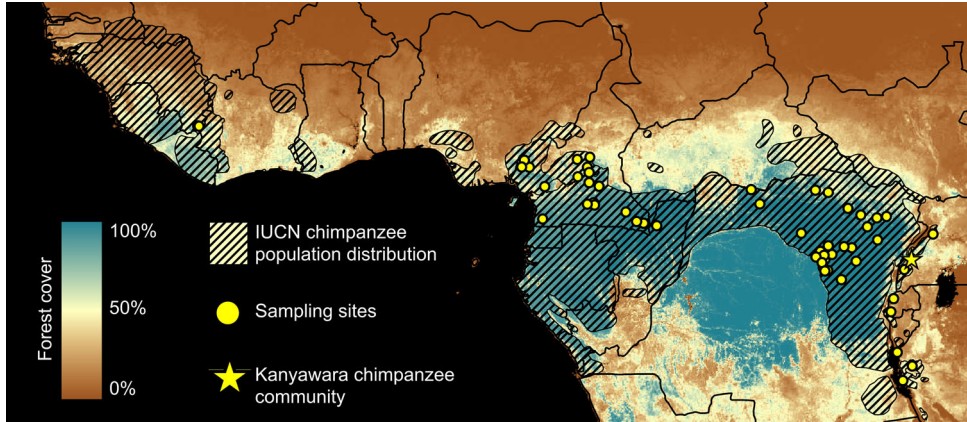

**Fig. 1 Geographic location of wild chimpanzee sampling sites.** Study sites are shown in relation to the geographic range of chimpanzees (*Pan troglodytes*). A total of N = 3314 chimpanzee fecal samples were analyzed in this study. N = 878 fecal samples were collected longitudinally from 54 members of the Kanyawara chimpanzee community in Kibale National Park, Uganda (yellow star). N = 2436 additional fecal samples were collected from 55 sampling sites across equatorial Africa (yellow circles). The coloration of the map corresponds to spatial variation in the percentage of forest cover (derived from ref. [62]).

temperature drives spatiotemporal variation in *Laverania* infection probability among chimpanzees. Infection probability peaked at ~24.5 °C among chimpanzee populations, consistent with the empirical transmission optimum of *P. falciparum*. Forest cover was also positively correlated with spatial variation in *Laverania* prevalence, consistent with the observation that forest-dwelling Anophelines constitute their primary vectors. Finally, we extrapolated these relationships across equatorial Africa to define spatiotemporal variation in the suitability of chimpanzee habitat for *Laverania* transmission, offering a hypothetical baseline indicator of human zoonotic exposure risk.

## Results

**Characterization of malaria infection within the Kanyawara chimpanzee community.** Malaria prevalence varies both spatially and temporally among wild chimpanzee populations[7,25], but no analysis has elucidated the ecological underpinnings of these dynamics. To investigate the epidemiology of *Laverania* infection in wild chimpanzees, we first employed a high-density, longitudinal sampling strategy. Between June 2013 and August 2016, we collected 878 fecal samples from the Kanyawara chimpanzee community living in Kibale National Park, western Uganda ($N = 1–29$ samples per individual; median = 18 samples; Supplementary Table 1). This community of wild chimpanzees— which has been under continuous observation since 1987[42]—is habituated to human observation, and all sampled inhabitants are identifiable by both morphology and by microsatellite genotype[43]. Samples were collected from 32 female ($N = 496$ samples) and 22 male ($N = 382$ samples) chimpanzees and stored in RNAlater at $-20$ °C until DNA extraction. At the time of sampling, subjects ranged in age between 0.3 and an estimated 55.8 years of age (median age at sampling: 16.6 years; Supplementary Table 1). Fecal samples were collected under direct observation and only when a positive identification was achievable. To evaluate identification reliability, DNA extracted from a subset of fecal samples ($N = 44$) was genotyped at 19 nuclear microsatellite loci, as described previously[44]. Microsatellite genotyping was successful in 97.7% of these samples (43/44), suggesting that fecal DNA was of sufficient quality for subsequent molecular analyses. Of the successfully genotyped samples, we observed a misidentification rate of 2.3% (1/43), indicating that identification fidelity was sufficient for analysis of demographic predictors of infection.

To quantify variation in *Plasmodium* infection within and among the Kanyawara chimpanzees, DNA was extracted from fecal samples and screened for malaria parasites using a single genome amplification (SGA) strategy via nested PCR, which targeted a 956-bp segment of the apicomplexan cytochrome B (cytB) mitochondrial gene as described previously[3,7]. To increase the sensitivity of this approach, we adopted an intensified PCR protocol, whereby all samples were screened in eight independent PCR reactions, as previously described[4]. All positive reactions were confirmed via direct sequencing without interim cloning. Of these 878 samples, 31.1% ($N = 217$) tested positive for one or more malaria parasite species in at least one replicate. Because the sensitivity of this assay to detect parasite DNA has been shown in humans to be greater in blood than in matched fecal samples, this value likely underestimates the frequency of blood-stage infection[18]. However, Loy et al.[18] also revealed that human fecal samples that tested positive by this approach exhibited a 26-fold increase in parasite DNA copy number in the blood relative to fecal samples that tested negative. Thus, it is likely that the positive fecal samples identified in our study reflect chimpanzees that harbor elevated parasitemia, which is a correlate of both morbidity[45–47] and transmission potential[48,49], at least in humans.

To diagnose the parasite species responsible for each infection, we aligned the newly generated sequences to a set of previously published cytB reference sequences (Supplementary Data 1) and constructed a Bayesian phylogenetic tree using MrBayes (version 3.2.6)[50] (Fig. 2a). Parasite species identity was confirmed via NCBI nucleotide BLAST. As expected, the majority of positive samples were attributable to chimpanzee *Laverania* parasites (23.6% *P. gaboni*; 8.0% *P. reichenowi*; 4.9% *P. billcollinsi*; Fig. 2b). Two additional members of the primate malaria clade were amplified in a minority of samples (1.2% *P. vivax*-like; 0.1% *P. malariae*-like; Fig. 2b), and 6.0% of samples were found to harbor multiple parasite species. These results demonstrate that the Kanyawara chimpanzees carry an assemblage of malaria parasites that is largely representative of the parasite diversity in chimpanzees across equatorial Africa[2,7,9,51].

**Evolutionary relationships of chimpanzee Hepatocystis parasites.** In addition to the *Plasmodium* species highlighted above, three samples (0.4%) harbored parasite sequences attributable to *Hepatocystis* spp., a clade of mammalian malaria parasites primarily isolated from Old World monkeys and bats[1,41,52]. Despite its divergent nomenclature, the *Hepatocystis* clade is more closely related to the primate malaria clade (e.g., *P. vivax*, *P. malariae*, *P. ovale*) than either is to *Laverania*[41,53]. Previous studies have similarly identified *Hepatocystis* in chimpanzee fecal samples[14]. However, to our knowledge, no study has evaluated the evolutionary relationships of these parasites relative to the large diversity of previously described *Hepatocystis* lineages to evaluate whether these species constitute (A) a novel *Hepatocystis* clade endemic to chimpanzee hosts or (B) the product of multiple independent cross-species transmission events originating in other mammals.

To discriminate between these possibilities, we analyzed the phylogenetic relationships of 20 great ape *Hepatocystis* sequences (i.e., three Kanyawara-derived sequences and 17 unpublished sequences amplified fortuitously during previous molecular epidemiological studies of ape malaria parasites[1,2,7]) relative to 56 previously published *Hepatocystis* cytB sequences (Supplementary Data 2) isolated from a wide range of hosts, including African and Asian monkeys, bats, and other mammals (Fig. 3). To determine the species origin of the ape fecal samples, we sequenced host mitochondrial D loop fragments and discarded sequences from samples from which this gene did not amplify. Of the 17 unpublished sequences, 15 were derived from chimpanzee fecal samples, while one was amplified from a dried blood spot (LIpts50094), collected from a captive chimpanzee living in the Limbe Wildlife Centre near Douala, Cameroon (Supplementary Data 3). Still another fecal sample contained human mtDNA sequences. The human-derived sample was collected at the Bafwabula field site in the Democratic Republic of the Congo in January 2007. While this would be the first report of a *Hepatocystis* infection in a human, we cannot exclude the presence of low-level primate DNA in the fecal sample, potentially resulting from the consumption of monkey bushmeat as the source of the *Hepatocystis* amplicon.

These phylogenetic analyses revealed that all ape-derived *Hepatocystis* isolates clustered within a clade of *Hepatocystis* lineages derived from African monkey hosts, and no isolate clustered with sequences from Asian monkeys, bats, or other mammals (Fig. 3). These ape lineages were phylogenetically interspersed among African monkey isolates. This pattern contrasts starkly with the topology of the *Laverania* phylogeny, which consists of largely host-specific clades despite extensive sympatry of hosts. Based on these data, it is possible that the great ape-derived *Hepatocystis* lineages are the result of multiple

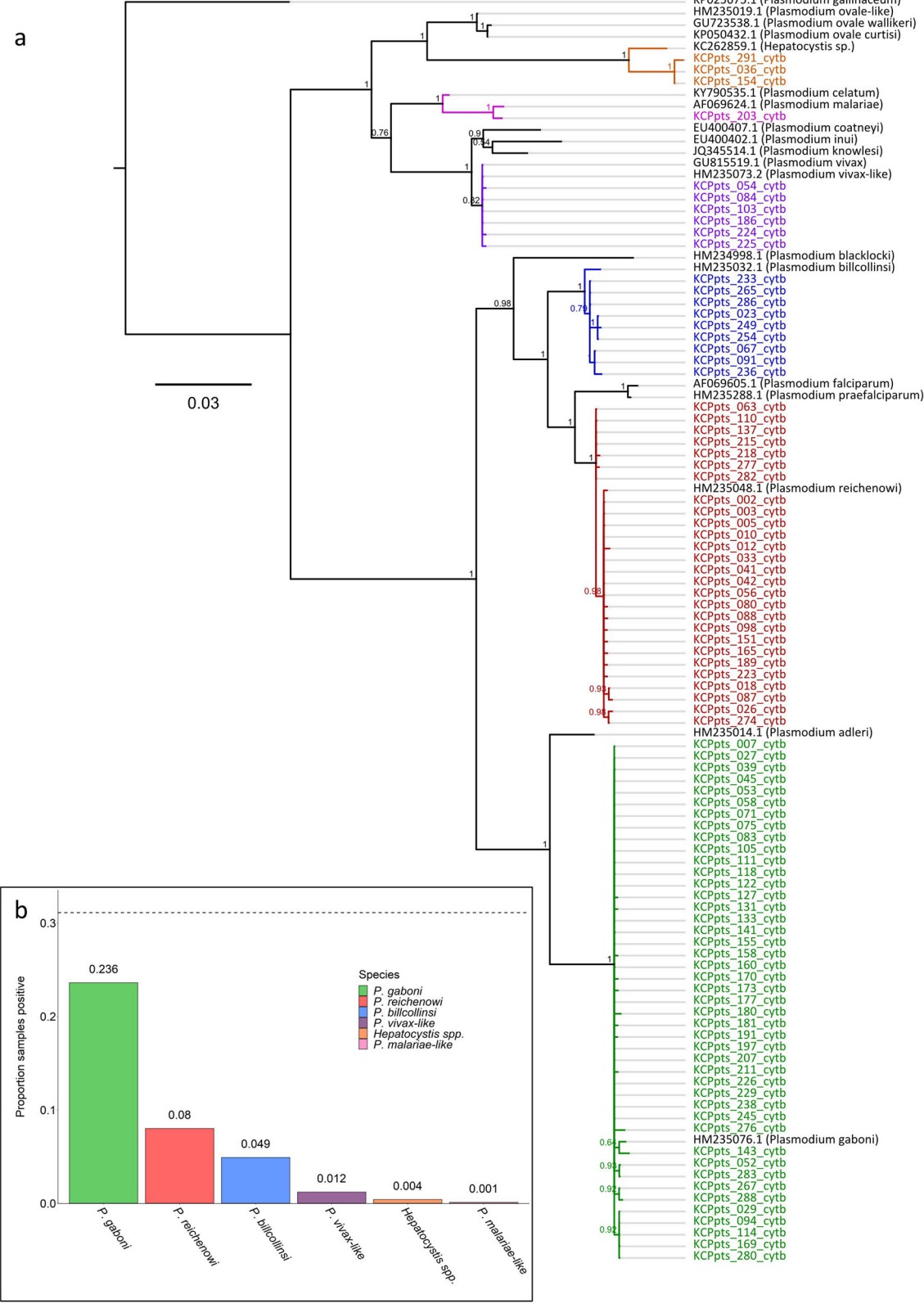

independent cross-species transmission events, suggesting that these parasites may cross species barriers promiscuously[54]. However, it is also possible that some, or all, of the fecal-derived chimpanzee *Hepatocystis* sequences resulted from the consumption of *Hepatocystis*-infected monkeys. Nevertheless, the fact that one *Hepatocystis* sequence was derived from a chimpanzee blood sample indicates productive infection in at least some cases.

**Ecological determinants of Laverania infection among the Kanyawara chimpanzees.** Given the high prevalence of *Laverania* infection harbored by the Kanyawara chimpanzees (Fig. 2b), we evaluated the extent to which the incidence of these parasites varied temporally among chimpanzee hosts. Between November and May, the proportion of positive samples was greater than the dataset mean (31.1%) in 9 out of 11 sampling months, and that

**Fig. 2 Richness of malaria parasites isolated from the Kanyawara chimpanzee community. a** Bayesian phylogeny generated from a representative subset of the SGA-derived cytB sequences generated in this study. Sequences were derived from $N = 878$ fecal samples, collected from the Kanyawara chimpanzee community in Kibale National Park, western Uganda, between 2013 and 2016. Color corresponds to malaria parasite species, inferred from the phylogenetic relationships of sequences to previously published reference sequences (green: *P. gaboni*, red: *P. reichenowi*, blue: *P. billcollinsi*, violet: *P. vivax*-like, orange: *Hepatocystis* spp., magenta: *P. malariae*). Newly generated sequences are listed in Supplementary Data 3, and previously published reference sequences used in this figure are listed in Supplementary Data 1. **b** Distribution of parasite species isolated from the Kanyawara chimpanzees. Malaria parasites were amplified in 31.1% of samples (dotted line). A majority of parasites amplified were members of the sub-genus *Laverania* (i.e., relatives of *P. falciparum*): *P. gaboni* (23.6%), *P. reichenowi* (8.0%), and *P. billcollinsi* (4.9%). Members of the primate malaria clade were amplified in a minority of samples: *P. vivax*-like (1.2%) and *P. malariae*-like (0.1%). In addition, *Hepatocystis* was amplified from 0.4% of samples.

samples were more likely to test positive for malaria parasites during this part of the year ($N = 404$ samples; positive samples expected $= 125.6 \pm 9.3$; positive samples observed $= 163$; binomial test, $p < 0.0001$; Fig. 4a). By contrast, the proportion of samples that tested positive was greater than the dataset mean in only 3 out of 13 sampling months between June and October, and samples were generally less likely to test positive for malaria parasites during these months ($N = 474$ samples; positive samples expected $= 147.4 \pm 10.1$; positive samples observed $= 110$; binomial test, $p < 0.0001$; Fig. 4a). Taken together, these results highlight a seasonal pattern of infection.

To evaluate the ecological drivers of infection probability, daily measurements of ambient temperature (minimum, maximum) and rainfall were collected from Kanyawara during the study period. We defined mean ambient temperature as the average of minimum and maximum air temperature estimates recorded during the 30 days prior to sample collection and defined intra-day temperature variation as the average difference between minimum and maximum temperature estimates during the same period. Mean ambient temperature during sample collection ranged from 20.0 to 23.0 °C (median: 20.7 °C) and intra-day temperature variation ranged from 7.9 to 15.6 °C (median: 11.0 °C). Mean daily rainfall during the 30 days prior to sample collection ranged from 0 to 14.5 mm per day (median: 3.0 mm).

To discern the ecological drivers of this seasonal pattern of infection, we used a generalized linear mixed model (GLMM)[55] with binomial error structure and logit link function to evaluate the relationship between climatic variation and the probability of *Plasmodium* infection among the Kanyawara chimpanzees. For each sample, we specified malaria infection (binary) as the outcome variable, mean ambient temperature (quadratic) and rainfall as fixed effects, chimpanzee individual as a random effect, and age (quadratic), sex, and intra-day temperature variation as covariates (Table 2). A full model that included all predictor variables and covariates provided a more parsimonious fit to the data than did a null model containing only random effects (likelihood ratio test: $\chi^2 = 64.5$, d.f. $= 6$, $p < 0.0001$; $\Delta$AIC $= 52.5$). As predicted, ambient temperature was positively correlated with the probability of infection, though the quadratic term had no effect and was omitted from the best-fit model (Fig. 4c; Table 2). Unexpectedly, we found that rainfall was negatively correlated with infection probability (Table 2). These results highlight ambient temperature as an important climatic predictor of seasonal variation in chimpanzee *Plasmodium* infection probability for this community.

Demographic variables also influenced infection probability. As reported previously[13], fecal samples of younger chimpanzees (including the youngest chimpanzee in the dataset, 3.7 months old at time of sampling) were more likely to test positive for malaria parasites than were older individuals (Fig. 4b, d; Table 2). This finding is consistent with the epidemiology of *P. falciparum* in humans, in which children under five years of age experience the bulk of morbidity and mortality before gradually developing immunity to the parasite upon repeated exposure[56–58]. Interestingly, however, while most chimpanzees adhered to this pattern,

some notable outliers were evident. For example, the fourth youngest chimpanzee in the dataset (NT; mean age: 1.28 years; Fig. 4b) tested positive for *Plasmodium* infection in 0/12 samples (model prediction: $7.6 \pm 2.8$ positive samples; $p < 0.0001$). Although the father of this chimpanzee (OG) exhibited an unremarkable pattern of infection (positive in 6/15 samples), her mother (NP) tested positive for *Plasmodium* in 1/29 samples, and her maternal grandfather (LK) was positive in 0/25 samples (Fig. 4b). It remains unclear whether such striking outliers are completely refractory to infection or are capable of controlling parasitemia such that it is below the limit of detection in feces. Future studies will be necessary to evaluate whether this pattern is attributable to genetic (e.g., erythrocyte receptor polymorphism) or other factors.

Taken together, these results demonstrate that the epidemiology of chimpanzee *Laverania* infection is characterized by high prevalence, early age of onset, and seasonality of infection, consistent with an elevated magnitude of active transmission, as opposed to rare, exclusively chronic infection. Given the promiscuous biting preferences of ape malaria vectors[17], these results provide complementary, albeit indirect, support for the assertion that humans living on the borders of chimpanzee habitat with active transmission may be exposed to their parasites and that the apparent host-specificity of the *Laverania* may be more attributable to cellular—rather than ecological—barriers to infection.

**Ecological analysis of spatiotemporal variation in malaria among chimpanzees across equatorial Africa.** To evaluate the generalizability of our longitudinal results of the Kanyawara chimpanzee community, we analyzed 2436 additional fecal samples collected between November 2000 and May 2016 from 55 sampling sites distributed across wild chimpanzee habitat in equatorial Africa ($N = 1$–258 samples per site; median $= 29$; Fig. 1), including both previously published ($N = 1936$) and unpublished ($N = 500$) samples (Table 1). As above, samples were screened for parasite mtDNA using a SGA strategy targeting a 956-bp cytB amplicon, as described previously[3].

Although direct measurements of climate were not available from all sites, satellite remote sensing technology can estimate climatic variables[59]. To estimate ambient temperature at each sampling site, we extracted daily Land Surface Temperature (LST) estimates from the MODerate Resolution Imaging Spectroradiometer (MODIS) thermal sensor on board the NASA-Terra satellite at 1 km$^2$ (downsampled to 10 km$^2$) spatial and 1-day temporal resolution (MOD11A1v005; http://modis.gsfc.nasa.gov/)[60]. We derived minimum and maximum air temperature estimates by applying the transformation outlined in Weiss et al.[36]. As above, mean ambient temperature was defined as the average of minimum and maximum air temperature estimates recorded during the 30 days prior to sample collection and intra-day temperature variation as the average difference between minimum and maximum temperature estimates during the same period. Rainfall estimates were derived from the Global Precipitation Climatology Project (GPCP V2.3; https://www.esrl.noaa.gov/psd/)[61] at 1-day temporal resolution and

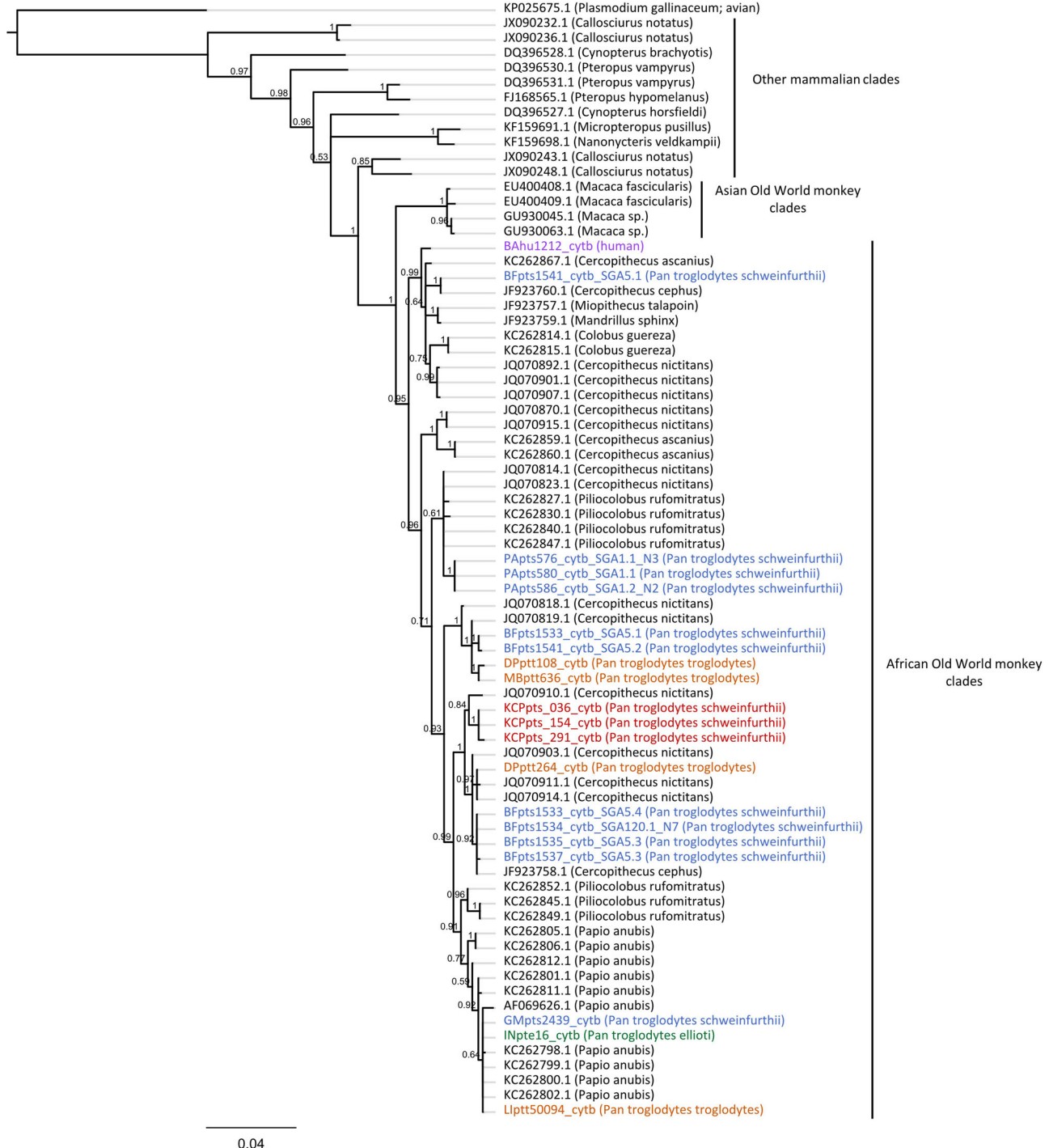

**Fig. 3 Phylogeny of *Hepatocystis* parasites isolated from chimpanzees and one putative human host.** Bayesian phylogeny generated from an alignment of *N* = 19 SGA-derived *Hepatocystis* cytB sequences (965 bp) produced in this study and *N* = 56 previously published *Hepatocystis* sequences from African Old World monkeys, Asian Old World monkeys, and other mammals. Chimpanzee parasite lineages cluster within a wide range of African Old World monkey parasites, suggesting a capacity to cross species boundaries. Sequences are annotated with respect to mammalian host species. Tip labels corresponding to *Hepatocystis* lineages isolated from Kanyawara fecal samples are colored red, and one human sample is colored violet. The remaining tip labels are colored with respect to chimpanzee sub-species (blue: *Pan troglodytes schweinfurthii*; orange: *Pan troglodytes troglodytes*; green: *Pan troglodytes ellioti*). Newly generated sequences are listed in Supplementary Data 3, and previously published reference sequences used in this figure are listed in Supplementary Data 2.

1-degree spatial resolution. Forest cover estimates were derived from high-resolution global maps of tree canopy cover (30 m, downsampled to 10 km), published by Hansen et al.[62]. The resulting dataset of 2436 samples included only those data points for which all predictor variables were available. Mean ambient temperature during the 30 days prior to sample collection ranged from 9.8 to 27.3 °C (median: 23.8 °C). Mean daily rainfall during the 30 days prior to sample collection ranged from 0 to 14.8 mm per day (median: 4.0 mm). Intra-day temperature variation during the 30 days prior to sample collection ranged from 6.0 to 13.4 °C

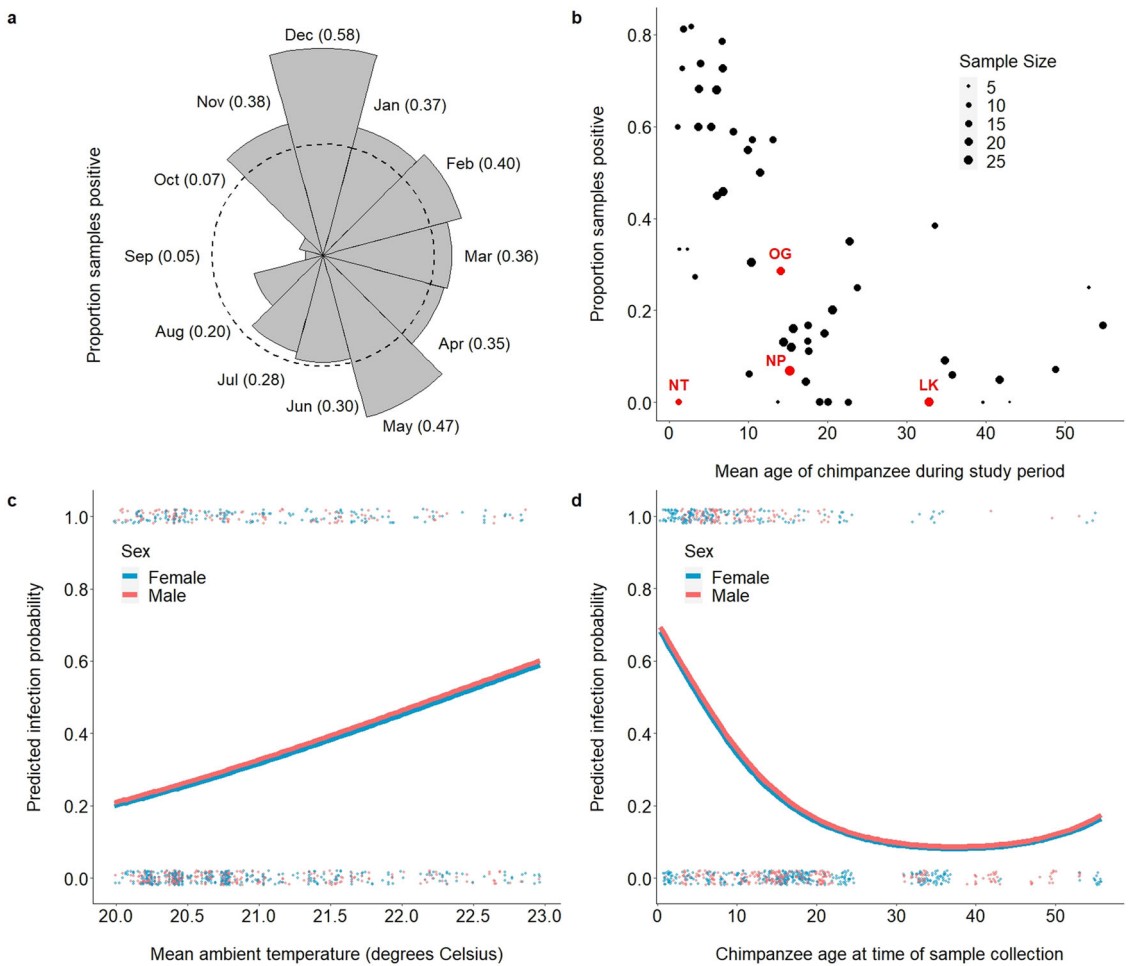

**Fig. 4 Longitudinal analysis of malaria parasitism in the Kanyawara cohort of wild chimpanzees. a** The proportion of chimpanzee fecal samples ($N = 878$) that tested positive for malaria parasites varied by month of sampling (dashed line corresponds to dataset mean). Bar length corresponds to proportion of samples that tested positive during a given month of sampling (monthly mean is listed in parentheses). Samples collected between November and May tended to be more likely to test positive for malaria parasites, while samples collected between June and October tended to be less likely. **b** Demographic variables also influenced infection probability. Samples collected from younger chimpanzees were generally more likely to test positive for malaria parasites than were samples collected from older chimpanzees. This result highlights the early age of infection onset (including the youngest sample in the dataset, collected from a 3-month-old female), indicative of a high magnitude of ongoing transmission. Despite this observation, a subset of chimpanzees (e.g., NT) deviated from this trend, potentially indicative of resistance to infection. **c** Predicted infection probabilities, derived from the Kanyawara GLMM (Table 2), demonstrate that seasonality of infection is partially driven by variation in mean ambient temperature. Mean ambient temperature (measured directly via weather monitoring stations) was positively correlated with infection probability across the range of temperature values observed at this sampling site (20.0–23.0 °C; $p < 0.0001$). Raw data (binary) are plotted as dots and stratified vertically for visualization. **d** Infection probabilities predicted by the Kanyawara GLMM also demonstrated that the youngest study subjects tended to be the most likely to test positive for malaria parasites ($p < 0.0001$). Raw data (binary) are plotted as dots and stratified vertically for visualization.

(median: 8.8 °C). Forest cover at sampling sites ranged from 16 to 100% (median: 90%).

We used a GLMM with binomial error structure and logit link function to evaluate the relationship between ecological variation and the probability of *Laverania* infection among wild chimpanzees across equatorial African sampling sites. We coded *Plasmodium* infection (binary) as the outcome variable, mean ambient temperature (quadratic), rainfall, and percent forest cover as fixed effects, sampling site as a random effect, and daily temperature variation and number of replicates (i.e., 1 or 8–10, depending upon whether samples were screened using the conventional SGA method[3] or the intensified method[4], respectively) as covariates (Supplementary Table 1). A full model that included all predictor variables and covariates provided a more parsimonious fit to the data than did a null model containing only random effects (likelihood ratio test: $\chi^2 = 37.9$, d.f. = 5, $p < 0.0001$; $\Delta AIC = 27.9$). As predicted, ambient temperature

was positively correlated with infection probability at low-temperature values and negatively correlated with infection at high-temperature values (Fig. 5a; Table 3). Infection probability peaked at ~24.5 °C, consistent with the ~25 °C transmission optimum of *P. falciparum*[27,33]. In addition, forest cover was positively correlated with infection probability (Fig. 5a; Table 3), while rainfall did not influence infection and was omitted from the best-fit model (Table 3). As expected, the sensitivity of the assay was enhanced by usage of the intensive PCR protocol (i.e., by screening samples in 8–10 replicates versus one replicate; Table 3).

Finally, we extrapolated these relationships between ecological variables and the probability of chimpanzee *Laverania* infection across equatorial Africa to identify hypothetical exposure hotspots. To characterize ecological variation across equatorial Africa, we derived composite rasters from mean ambient temperature and intra-day temperature variation measurements

**Table 1 Chimpanzee fecal samples analyzed in this study.**

| Field sites tested | Field sites | Field sites positive | Samples | Samples positive | References |
|---|---|---|---|---|---|
| *Longitudinal analysis* | | | | | |
| Kanyawara (KCP) | 1 | 1 | 878 | 273 | This study |
| *Pan-African analysis* | | | | | |
| BO, GI, GM, KB, KY, MH, NB, NY, UG | 9 | 1 | 500 | 6 | This study |
| AM, AN, AZ, BA, BB, BD, BF, BG, BI, BL, BQ, CP, DG, DP, EB, EK, EN, EP, GO, GT, IS, KA, KO, KS, LB, LH, LU, MB, MD, MF, MK, MP, MT, MU, ON, OP, PA, PO, SL, UB, VM, WA, WB, WE, WL, YW | 46 | 31 | 1936 | 390 | [1,2,7] |
| Total | 55 | 32 | 2436 | 396 | |
| Combined dataset | 56 | 33 | 3314 | 669 | |

Key: Amunyala (AM), Ango (AN), Azunu (AZ), Babingi (BI), Bafwaboli (BA), Bafwasende (BF), Belgique (BQ), Bondo-Bili (BD), Bongbola (BL), Bossou (BO), Boumba Bek (BB), Budongo (BG), Campo Ma'an (CP), Diang (DG), Doumo Pierre (DP), E'kom (EK), Ebo (EB), Engali (EN), Epulu (EP), Gishwati (GI), Goalougo Triangle (GT), Gombari (GO), Gombe (GM), Isiro (IS), Kabuka (KA), Kagwene (YW), Kibale Kanyawara (KCP), Kibale Ngogo (KB), Kisangani (KS), Kotakoli (KO), Kyambura Gorge (KY), Liabelem Highlands (LH), Lobéké (LB), Lubutu (LU), Mahale (MH), Makombe (MK), Mamfé (MF), Manbele (MB), Mbam et Djerem (MD), Metep (MP), Minta (MT), Munbgere (MU), Nimba (NB), Nyungwe (NY), Onga (ON), Opienge (OP), Parisi (PA), Poko (PO), Somalomo (SL), Ubangi (UB), Ugalla (UG), Vome (VM), Walengola (WL), Wamba (WB), Wanie-Rukula (WA), Wassa Emtse (WE).

recorded between March 2000 and February 2017 across the African continent at 1-degree spatial and 1-month temporal resolution. Using these composite temperature rasters and the Hansen et al.[62] forest cover dataset, we projected the predicted probabilities of chimpanzee *Laverania* infection (derived from the pan-African model) across the spatial extent of chimpanzee habitat (as defined by the International Union for the Conservation of Nature, IUCN; Fig. 5c). The resulting risk map indicates spatial variation in the suitability of chimpanzee habitat for *Laverania* transmission across all seasons during an average year. Stratification of these results by month of sampling highlights temporal variation in infection probability and predicts elevated transmission between January and May, as well as a low transmission season between June and September, for much of the region (Fig. 5b, d). This risk map also highlights spatial variation in the seasonality of transmission (e.g., high levels of sustained transmission in Central Africa, contrasting a more sharply seasonal pattern of infection in habitat toward the edges of the chimpanzee habitat range). Although inter-annual variation would be expected to cause deviations from these estimates, this risk map offers a general baseline prediction for follow-up studies of exposure and transmission.

## Discussion

Although infection of wild chimpanzees with *Laverania* parasites is well-documented, the ecological drivers of spatiotemporal variation in prevalence remain largely unexplored. Our analyses define the ecological niche of chimpanzee *Laverania* parasites and support the hypothesis that variation in mean ambient temperature determines infection dynamics. Extrapolation of these relationships across equatorial Africa offers a framework for the estimation of transmission intensity in chimpanzees and thereby helps quantify the risk of human exposure to these parasites.

Ambient temperature influences numerous elements of the *Plasmodium* life cycle, including the population dynamics of the Anopheline vector and the rate of parasite development[26–31]. Both longitudinal and cross-sectional sampling strategies independently support the conclusion that mean ambient temperature is correlated with the probability of *Laverania* infection (Figs. 4c and 5a, Table 2), which peaks at ~24.5 °C in wild chimpanzee populations, close to the ~25.0 °C empirical transmission optimum of *P. falciparum* in humans[27,33]. The consistency of this result is underscored by the fact that these two approaches use non-overlapping datasets and alternative methods of temperature measurement (Kanyawara: direct measurement; pan-African: remote sensing). In addition to mean ambient temperature, intra-day temperature variation—which has also been shown to influence the epidemiological

**Table 2 Longitudinal analysis of malaria parasitism among the Kanyawara chimpanzees.**

| Parameter[a] | Estimate | Std Err | z | *P*-value |
|---|---|---|---|---|
| Intercept | −13.2 | 3.17 | −4.17 | <0.0001 |
| *Ecological* | | | | |
| Mean ambient temperature | 0.834 | 0.174 | 4.80 | <0.0001 |
| (Mean ambient temperature)$^2$ | NS[b] | NS[b] | NS[b] | NS[b] |
| Intra-day temperature variation | −0.466 | 0.085 | −5.48 | <0.0001 |
| Precipitation | −0.096 | 0.035 | −2.75 | 0.006 |
| *Demographic* | | | | |
| Age | −26.5 | 4.63 | −5.73 | <0.0001 |
| (Age)$^2$ | 15.4 | 4.45 | 3.47 | 0.0005 |
| Sex (male) | 0.073 | 0.319 | 0.23 | 0.819 |

[a]Output of longitudinal GLMM corresponding to the probability ape malaria parasite infection relative to ecological and demographic predictor variables of interest. Model based upon 878 fecal samples collected longitudinally from 54 individual wild chimpanzees of the Kanyawara cohort in western Uganda. Note that predictor variables were scaled to augment model convergence. All samples in the longitudinal analysis of the Kanyawara cohort were screened eight times for malaria parasites using an intensified SGA methodology, as described previously[3,4,7]. See "Methods" for model specification details.
[b]*Mean ambient temperature*$^2$ did not improve the fit of the longitudinal GLMM and was, therefore, omitted from the final, most parsimonious, model.

processes at the parasite-mosquito interface[28,37]—was correlated with infection probability in both datasets (Tables 2 and 3). However, the direction of this correlation differed in these analyses, which suggests a more complicated relationship with malaria infection (e.g., countervailing effects of temperature fluctuation when observed at mean ambient temperature extremes[28] or vector-specific effects[37]). Taken together, these results indicate that the temperature-sensitive epidemiology of the chimpanzee *Laverania* closely resembles that of *P. falciparum*, suggesting that conserved mechanisms govern the epidemiology of these taxa.

Forest cover was also positively correlated with infection probability (Fig. 5a and Table 3) which is consistent with the finding that forest-dwelling Anopheline species (*An. vinckei*, *An. moucheti*) constitute the primary vectors of these parasites[16,17] and supports a recent analysis of malaria parasitism across non-human primates[39]. Although these signatures of forest-dependent transmission suggest that human encroachment into undisturbed habitat may pose a greater exposure risk than does habitat fragmentation, these relationships are likely complex[40] and warrant further investigation at fine-grained spatial scales.

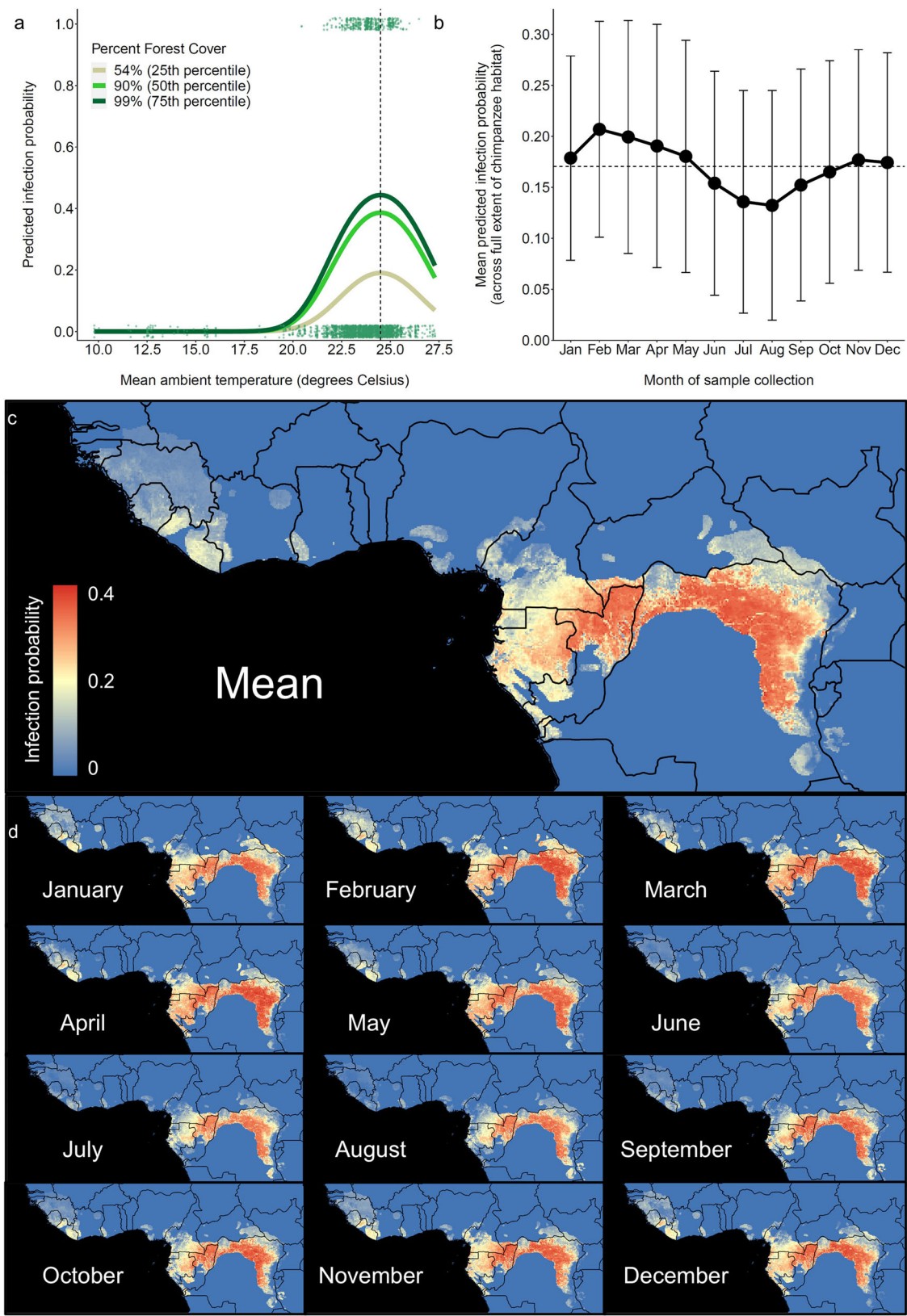

Extrapolation of these relationships across equatorial Africa highlights spatial and temporal variation in the ecological suitability of chimpanzee habitat for *Laverania* transmission (Fig. 5b–d). Although we found two distinct transmission seasons: (1) a "high season" between January and May, and (2) a "low season" between June and September, the amplitude and character of this seasonal pattern likely varies regionally. For example, northern Democratic Republic of the Congo and northern Republic of the Congo are predicted to support high levels of transmission across seasons, while Cameroon and Gabon will have infection probability dropping between June and August. Still other countries, such as Rwanda and Tanzania, are

**Fig. 5 Ecological niche modeling of malaria parasitism in wild chimpanzee reservoirs across equatorial Africa. a** Analysis of $N = 2436$ wild chimpanzee fecal samples collected from 55 sampling sites across equatorial Africa demonstrate that mean ambient temperature (inferred from MODIS remote sensing datasets; see Methods) and forest cover[62] are critical determinants of chimpanzee *Laverania* epidemiology. The pan-African GLMM (Table 3) demonstrates that (1) infection probability peaks ~24.5 °C (dotted vertical line), consistent with the empirical transmission optimum of *P. falciparum*[27], and (2) forest cover is positively correlated with infection probability ($p = 0.001$), consistent with the observation that forest-dwelling Anophelines constitute the primary vectors of these parasites[16,17]. Color corresponds to three categories of forest cover: 54% (25th percentile), 90% (50th percentile), and 99% (75th percentile). Raw data (binary) are plotted as dots and stratified vertically for visualization. **b**, **c** Given the ecological relationships identified in this study, we extrapolated infection probabilities across chimpanzee habitat in equatorial Africa. We derived composite rasters from mean ambient temperature and intra-day temperature variation measurements recorded between 2000 and 2017 across the African continent. Using these composite temperature rasters and the Hansen et al.[62] forest cover dataset, we projected the predicted probabilities of chimpanzee *Laverania* infection (derived from the pan-African model) across the spatial extent of chimpanzee habitat. The monthly mean pixel value of each raster is plotted, and error bars correspond to standard deviation of pixel values on each raster. **d** Stratification by month highlights the seasonality of these infections. Across chimpanzee habitat, infection probability peaks between the months of January and May, and infection probability declines between June and September. Because mosquito vectors of ape malaria parasites readily bite humans[17], these maps can serve as a baseline proxy for spatiotemporal variation in the risk of human exposure in areas where humans and apes overlap.

**Table 3 Ecological modeling of malaria parasitism among chimpanzee hosts.**

| Parameter[a] | Estimate | Std Err | z | P-value |
|---|---|---|---|---|
| Intercept | −2.54 | 0.338 | −7.50 | <0.0001 |
| Ecological | | | | |
| Mean ambient temperature | 0.355 | 0.127 | 2.79 | 0.005 |
| (Mean ambient temperature)$^2$ | −0.164 | 0.063 | −2.60 | 0.009 |
| Intra-day temperature variation | 0.208 | 0.099 | 2.09 | 0.036 |
| Precipitation | NS[b] | NS[b] | NS[b] | NS[b] |
| Percent forest cover | 0.032 | 0.010 | 3.20 | 0.001 |
| Technical | | | | |
| Number of replicates (8)[c] | 1.164 | 0.500 | 2.33 | 0.020 |

[a]Output of pan-African GLMM corresponding to the probability ape malaria parasite infection relative to ecological predictor variables of interest. Model was based upon 2436 fecal samples collected at 55 sampling sites in wild chimpanzee habitat across equatorial Africa. Note that predictor variables were scaled to augment model convergence. See "Methods" for model specification details.
[b]*Precipitation* did not improve the fit of the longitudinal GLMM and was, therefore, omitted from the final, most parsimonious, model.
[c]Samples in the pan-African analysis were screened either in either one or eight replicates.

predicted to harbor very low transmission intensity, a projection supported by the absence of infection in samples collected from these countries. Although these patterns are expected to vary between years as a function of local climatic anomalies, this map offers a baseline set of predictions for longitudinal follow-up studies of *Laverania* epidemiology.

In addition to highlighting variation in *Laverania* infection among wild chimpanzees, these maps offer a rough proxy for the estimation of human exposure risk. Our analyses suggest that the epidemiology of chimpanzee *Laverania* infection is defined by (1) high prevalence (Fig. 2), (2) early age of onset (Fig. 4b, d), and (3) seasonality of infection (Figs. 4a, c and 5a–d). These results are consistent with a high magnitude of ongoing transmission among chimpanzee hosts. In addition, human landing capture experiments conducted in Gabon have demonstrated that the Anopheline vectors of ape malaria parasites readily bite humans when given the opportunity[17]. Given that fecal parasite rates are indicative of blood stage parasitemia[18], which correlates with transmission potential[48,49], these observations suggest that the per capita probability of human exposure is likely to be greatest in areas where the magnitude of transmission among chimpanzees is similarly elevated. Accordingly, these habitat suitability maps highlight predicted human exposure hotspots, which can inform the design of zoonotic surveillance efforts that quantify exposure directly (e.g., via the measurement of entomological inoculation rates and serological exposure). Future studies should prioritize rural human populations living near forests containing Plasmodium-infected apes demonstrating active transmission in the Republic of Congo and the Democratic Republic of the Congo, as well as parts of southern Cameroon and eastern Gabon, between the months of January and May. Since gorillas are

infected with the precursor of *P. falciparum*, this ape species in particular should be included in epidemiological studies. Future efforts that fine-tune and extend this risk map (e.g., via the inclusion of human population density, ape population density, suitable mosquito vectors, and range overlap data layers) are necessary to operationalize these predictions.

Although it is likely that human exposure to chimpanzee malaria parasites occurs with regularity in parts of equatorial Africa, chimpanzee-to-human *Laverania* transmission has never been documented. Strong molecular barriers to cross-species *Laverania* transmission have been identified[19,20,22] and two surveys of rural human populations in Gabon and Cameroon[10,11], as well as an extensive survey of indigenous hunter gatherers in Cameroon[18], have failed to reveal evidence of zoonotic transmission. The strong host-specificity of the *Laverania* species contrasts with the lack of host-specificity of *Hepatocystis* parasites (Fig. 3). Though *Laverania* and *Hepatocystis* parasites each achieve high prevalence in hosts that occupy sympatric distributions in equatorial Africa, cross-species transmission of the former has not yet been convincingly documented[23,51,52]. While *Hepatocystis* parasites are known to cross species barriers rather promiscuously[54], our study suggests that chimpanzees are susceptible to monkey *Hepatocystis* infection (Fig. 3). Although some, or all, *Hepatocystis* sequences derived from chimpanzee fecal samples may have resulted from the consumption of *Hepatocystis*-infected monkeys, we also amplified one *Hepatocystis* sequence from the blood of a captive chimpanzee, providing evidence for a productive infection. This is the first description of a chimpanzee *Hepatocystis* infection and suggests that cross-species transmission of *Hepatocystis* between monkeys and chimpanzees (and potentially humans) may have been

underestimated. Future studies that screen human and chimpanzee populations for this parasite using *Hepatocystis*-specific primers will be necessary to examine the frequency of productive cross-species transmission.

The analyses presented here may also catalyze future studies of host-parasite coevolution in wild chimpanzee hosts. Although malaria pathogenesis ranks amongst one of the strongest selective pressures to confront human populations during the past 50,000 years[63,64], little is known about the pathogenicity of the *Laverania* in chimpanzees. While some case studies have documented several characteristic symptoms—such as fever, anemia, elevated parasitemia, and possibly mortality—in *Plasmodium* unexposed captive chimpanzees upon initial exposure to these parasites[45,65], other studies have produced equivocal results[66]. If *Laverania* parasites are routinely pathogenic in wild chimpanzees, natural selection could plausibly have favored the evolution of mechanisms of resistance. Three closely related individuals in our sample had exceptionally high frequencies of testing negative for malaria parasites, raising the possibility that they shared a genetic polymorphism conferring resistance. It is also conceivable that chimpanzees have evolved a series of behavioral responses (e.g., medicinal plant use[67]) to mitigate malaria pathogenesis. Future studies that synthesize molecular and behavior approaches at long-term study sites, such as Kanyawara, will facilitate investigation of these hypotheses.

Although efforts to control malaria have achieved considerable success during recent decades, the emergence of non-human primate malaria parasites in Southeast Asia (*P. knowlesi*[68,69]) and South America (*P. simium*[70]) have raised concerns that zoonoses could rollback these hard-fought gains. While the ape *Laverania* appear to exhibit considerable host-specificity on contemporary timescales, there is nevertheless a strong impetus to critically evaluate the zoonotic potential of these parasites. Chimpanzees and gorillas live in close proximity to humans in parts of equatorial Africa, two pandemic malaria parasites (*P. falciparum*, *P. vivax*) have emerged from African great ape reservoirs in the past[1,7], and evidence of contemporary human-to-ape transmission of *P. falciparum* has been documented[23,51,71,72]. In addition, chimpanzee and gorilla *Laverania* parasites have transcended primate species barriers in captivity and vectors of ape malaria parasites exhibit promiscuous biting preferences[17]. Accordingly, it is likely that human exposure occurs with regularity in parts of equatorial Africa. Given finite resources, it is critical that any effort to evaluate the zoonotic potential of the chimpanzee *Laverania* adopt a targeted approach. Our results offer an ecological framework that will facilitate the design of surveillance efforts by highlighting where and when human exposure is most likely to occur.

## Methods

**Chimpanzee samples.** In this study, we used a combination of longitudinal and cross-sectional sampling strategies to collect 3314 fecal samples from wild chimpanzees for molecular analyses of malaria parasitism. For longitudinal analyses, 878 fecal samples were collected from 54 chimpanzees inhabiting the Kanyawara chimpanzee community, located in Kibale National Park, western Uganda (Fig. 1; Supplementary Table 1). This cohort of wild chimpanzees was habituated to human observation by RWW and has been under continuous direct observation since 1987. All Kanyawara study subjects are identifiable by both morphological appearance and microsatellite genotype. At this field site, researchers and field assistants conduct focal follows of individual chimpanzees on a daily basis and record individual-level behavior, party-level behavior, social affiliation data, and biological samples, among other parameters. Fecal samples—collected only upon direct observation of defecation and only when a positive identification was certain—were preserved in RNA*later* (1:1 vol/vol) and stored at −20 °C until exportation and DNA extraction. For cross-sectional analyses, 2436 chimpanzee fecal samples were collected from 55 chimpanzee field sites, distributed across equatorial Africa (Fig. 1; Supplementary Table 2). This dataset included 1936 wild chimpanzee samples, previously collected for molecular studies of simian retroviruses[73–78] or malaria parasites[1,7], and 500 samples that were newly collected for these analyses

(Table 1). Samples were only included in this dataset if collection dates were recorded and corresponding ecological variables (see below) were available. This cross-sectional dataset comprised a combination of samples that were either collected from habituated chimpanzees occupying long-term research sites or opportunistically from non-habituated chimpanzees during ape and biodiversity surveys. Samples were collected in RNA*later* (1:1 vol/vol), transported at ambient temperature, and stored at −80 °C. For both longitudinal and cross-sectional surveys, DNA was extracted using the QIAamp Stool DNA minikit (Qiagen, Valencia, CA). Confirmation of host species was obtained either by direct observation of defecation (longitudinal analyses) or via amplification and sequencing of host mitochondrial DNA[73–78] (cross-sectional analyses).

Molecular epidemiological studies of wild-living chimpanzees at the Bafwaboli (BA) field site also yielded a fecal sample from an unknown human, which was identified by mtDNA analysis. This human fecal sample as well as a dried blood spot sample collected from a captive chimpanzee (50094) at the Limbe Wildlife Centre in Cameroon were both positive for *Hepatocystis* mtDNA sequences (965 bp) when screened for *Laverania* infections using cross-reactive PCR primers (Supplementary Data 3).

**Microsatellite analyses.** To evaluate the accuracy of Kanyawara chimpanzee identification, we used a semi-nested multiplex PCR protocol, as described previously[43,44] (with slight modifications), to genotype 44 fecal samples at 19 polymorphic microsatellite loci. Briefly, all microsatellite loci were first amplified in tandem via an initial multiplexing step. For each sample, 5 μL of extracted DNA was added to a 20 μL multiplex PCR reaction containing all 19 primer pairs at 0.15 mM concentration, 1X Expand Long Template Buffer without MgCl₂, 1.75 mM MgCl₂, 110 μM of each dNTP, 16 μg bovine serum albumin (BSA), and 0.5 U of Expand Long Template Taq DNA polymerase. Thermocycling was performed with an initial denaturation step for 5 min at 95 °C, 30 cycles of 30 s at 95 °C, 90 s at 58 °C, and 30 s at 72 °C, followed by a final extension at 72 °C for 30 min. Next, a singleplex PCR was performed for each locus in 10 μL volume by mixing 5 μL of the multiplex reaction (diluted 1:100), 0.25 mM of the corresponding fluorescently labeled (FAM, HEX, or NED) forward primer, 0.25 mM nested reverse primer, 1X Expand Long Template Buffer without MgCl₂, 0.875 mM MgCl₂, 110 μM of each dNTP, 8 μg BSA, and 0.25 U of Expand Long Template Taq DNA polymerase. Thermocycling was performed with an initial denaturation step for 5 min at 95 °C, 30 cycles of 30 s at 95 °C, 90 s at primer-specific annealing temperatures, and 30 s at 72 °C, followed by a final extension at 72 °C for 30 min. Amplification products were electrophoresed on an ABI PRISM 3100 Genetic Analyzer and sized relative to an HD400 (ROX) size standard using GeneMapper 5.0 (Applied Biosystems). This procedure was repeated up to six times for loci that failed to amplify. The microsatellite genotype derived from each of 44 samples was then compared to those of the Kanyawara chimpanzees, and incorrectly attributed samples were recorded as such.

**Conventional PCR and single genome amplification.** To quantify variation in malaria infection, fecal DNA was screened for malaria parasites using a conventional nested PCR, targeting a 956-bp segment of the apicomplexan mitochondrial cytB gene, using external primers DW2 (5′-TAATGCCTAGACGTATTCCTGA TTATCCAG-3′) and DW4 (5′-TGTTTGC TTGGGGAGCTGTAATCATAATGT G-3′) in the first-round PCR, and internal primers Pfcytb1 (5′-CTCTATTAATT TAGTTAAAGCACA-3′) and PLAS2a (5′-GTGGTAATTGACATCCWATCC-3′) in the second round, as described previously[3,7]. To increase the sensitivity of this approach, a subset of fecal samples were subjected to an intensified PCR protocol, whereby samples were screened in 8–10 independent PCR reactions, as previously described[4]. All Kanyawara fecal samples and a subset of pan-African samples (Supplementary Data 3) were subjected to this intensified PCR protocol. To minimize PCR-induced errors, we applied a single genome amplification (SGA) approach to each fecal sample found to be positive by conventional or intensive PCR, as described previously[1,3,7]. Briefly, fecal DNA from positive samples was end point diluted in 96-well plates until <30% of wells tested positive. Given a Poisson distribution, this dilution will yield a single amplifiable template per positive reaction >80% of the time. All reactions that were identified to be positive using this SGA methodology were sequenced directly without interim cloning, yielding sequences that were derived from a single template.

**Phylogenetic analyses.** For analysis of malaria parasitism within the Kanyawara chimpanzee community, we trimmed sequences to 863 bps and used Geneious aligner (version 10) to align sequences to a set of phylogenetically informative reference sequences, corresponding to a representative diversity of simian and human malaria parasites. Sequences shorter than 863 bps were omitted from phylogenetic analyses. Alignments were then visually inspected and sequences containing ambiguities were removed from subsequent analyses. We used jModelTest (version 2.0)[79] to identify the best-fit nucleotide substitution model and MrBayes (version 3.2.6)[50] to generate Bayesian posterior probabilities, using a chain length of 5.5 million and 10% burn-in. Convergence was achieved when the average deviation of split frequencies was <0.01. Parasite species identity was inferred via NCBI nucleotide BLAST and confirmed using the phylogenetic relationships generated above. To analyze the evolutionary relationships of chimpanzee

*Hepatocystis* isolates, we used the methodology outlined above, except sequences were trimmed to 815 bps to allow for comparison to a greater diversity of *Hepatocystis* reference sequences. In this case, chimpanzee and human *Hepatocystis* isolates were aligned to a representative diversity of previously published *Hepatocystis* isolates, spanning parasites of African Old World monkeys, Asian Old World Monkeys, and other mammals.

**Estimation of ecological parameters**. We estimated ecological variables using two alternative methodologies: direct measurement and remote sensing. For longitudinal analyses of the Kanyawara chimpanzee community, CAC recorded measurements of ambient air temperature (minimum, maximum) and rainfall twice daily during the study period via local weather monitoring stations installed at the field site. Mean ambient temperature was calculated as the average of minimum and maximum ambient air temperature measurements and intra-day temperature variation was calculated as the difference between these values. For each sample, we defined mean ambient temperature, intra-day temperature variation, and rainfall as the average of measurements of the corresponding variable during the 30 days prior to sample collection. For cross-sectional analyses, ecological variables were estimated using remote sensing datasets. For each sample, we downloaded daytime and nighttime Land Surface Temperature (LST) measurements from the MODerate Resolution Imaging Spectroradiometer (MODIS) thermal sensor on board the NASA-Terra satellite at 1 km² spatial resolution and 1-day temporal resolution (MOD11A1v005; http://modis.gsfc.nasa.gov/)[60] at the geospatial point of sample collection. We subsequently performed quality control to remove low quality or missing data (e.g., resulting from cloud cover) and downsampled spatial resolution to 10 km² to increase applicability of variables across the sampling site. We then converted these daytime and nighttime LST estimates to minimum and maximum ambient temperature using the equations presented in Weiss et al.[36] (with corrected typographical error; Daniel Weiss, pers. comm.):

$$T_{min} = 0.209 + 0.971 \times LST_{night} \qquad (1)$$

$$T_{max} = 18.149 + 0.949 \times LST_{day} + -0.541 \times (LST_{day} - LST_{night})$$
$$+ -0.866 \times Day_{length} \qquad (2)$$

where $T_{min}$ is minimum ambient temperature, $T_{max}$ is maximum ambient temperature, $LST_{night}$ is the nighttime LST estimate, $LST_{day}$ is the daytime LST estimate, and $Day_{length}$ is the number of daylight hours on the day of sampling. Given these values, we calculated mean ambient temperature as the average of minimum and maximum ambient air temperature measurements and intra-day temperature variation was calculated as the difference between these values, as above. Rainfall estimates were derived from the Global Precipitation Climatology Project (GPCP V2.3)[61] at 1-day temporal resolution and 1-degree spatial resolution (https://www.esrl.noaa.gov/psd/). Again, for each sample, we defined mean ambient temperature, intra-day temperature variation, and rainfall as the average of measurements of the corresponding variable during the 30 days prior to sample collection. Forest cover estimates were derived from high-resolution global maps of tree canopy cover (30 m, downsampled to 10 km), published by Hansen et al.[62]. The resulting dataset included only those data points for which all predictor variables were available ($N = 2436$ samples).

**Statistics and reproducibility**. We developed two separate generalized linear mixed models to identify the ecological and demographic determinants of *Laverania* prevalence in wild chimpanzee hosts. For longitudinal analyses (i.e., the Kanyawara GLMM), we specified a generalized linear mixed model (GLMM)[55] with binomial error structure and logit link function to evaluate the relationship between climatic variation and the probability of *Plasmodium* infection among the Kanyawara chimpanzees. For each sample, we specified malaria infection (binary) as the outcome variable, mean ambient temperature (quadratic) and rainfall as fixed effects, chimpanzee individual as a random effect, and age (quadratic), sex, and intra-day temperature variation as covariates (Table 2). Similarly, for cross-sectional analyses (i.e., the pan-African GLMM), we specified a GLMM with binomial error structure and logit link function to evaluate the relationship between climatic variation and the probability of *Plasmodium* infection among wild chimpanzees inhabiting sampling sites across equatorial Africa. In this case, we coded *Plasmodium* infection (binary) as the outcome variable, mean ambient temperature (quadratic), rainfall, and percent forest cover as fixed effects, sampling site as a random effect, and daily temperature variation and number of replicates (i.e., 1 or 8–10, depending upon whether samples were screened using the conventional SGA method[3] or the intensified method[4], respectively) as covariates (Table 3). Variables in the Pan-African analysis were mean-centered to improve model stability. In each case, we used likelihood ratio tests to compare the fit of the full GLMM (containing all predictor variables, covariates, random effect, and intercept) to that of the null model (containing only random effect and intercept). Predictor variables that did not improve the fit of each model were subsequently excluded from the final models. These statistical analyses were performed in R (version 3.4.3) using the lme4 package[80].

**Geospatial mapping**. Given the ecological relationships identified in the Pan-African GLMM, outlined above, we extrapolated infection probabilities across

wild chimpanzee habitat in equatorial Africa. To characterize ecological variation across equatorial Africa, we derived composite rasters from mean ambient temperature and intra-day temperature variation measurements recorded between March 2000 and February 2017 across the African continent at 1-degree spatial and 1-month temporal resolution. Using these composite temperature rasters and the Hansen et al.[62] forest cover dataset, we projected the predicted probabilities of chimpanzee *Laverania* infection (derived from the pan-African model) across the IUCN-designated spatial extent of chimpanzee habitat, according to the following equation:

$$\text{Predicted probability} = \frac{1}{1 + e - (\text{intercept} + 0.355 \times AT - 0.164 \times AT^2 + 0.032 \times FC + 0.208 \times TV)} \qquad (3)$$

where intercept is −2.538 for samples screened using conventional PCR (i.e., one replicate) and −1.374 for those screened using intensive PCR (i.e., 8–10 replicates), and *AT*, *TV*, and *FC* are each corrected by subtracting the means of the chimpanzee dataset (23.4 for *AT*, 9.1 for *TV*, and 77.2 for *FC*). All geospatial analyses were performed in ArcMap (v10.6 Environmental Systems Research Institute Inc.).

**Reporting summary**. Further information on research design is available in the Nature Research Reporting Summary linked to this article.

## Data availability

*Plasmodium* and *Hepatocystis* mtDNA sequence generated for this study have been deposited at NCBI GenBank under the accession numbers MW228501-MW228790 and OL691962-OL691978, respectively (Supplementary Data 4). All other datasets generated in this study are available from the corresponding authors upon request.

## Code availability

All statistical analyses were performed using open-source packages in R (version 3.4.3) as referenced in the methods section. The scripts used to implement these analyses are available from the corresponding authors upon request.

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

## Acknowledgements
This work was supported by grants from the National Institutes of Health R01AI091595, R01AI120810, R01AI050529, and P30AI045008 (B.H.H.); R01HL139337 (M.T.D.), the National Geographic Society (E.J.S.), the International Primatological Society (E.J.S.), and the American Society of Primatologists (E.J.S.), as well as fellowships from Harvard University (E.J.S.) and the National Science Foundation (E.J.S.). We thank Dr. Kristen Skillman for help in putting the manuscript together, as well as two referees for their review and thoughtful comments on this manuscript.

## Author contributions
E.J.S. wrote the manuscript, generated all figures and tables, collected chimpanzee fecal samples from Kibale National Park, Budongo Forest, and Kyambura Gorge, extracted DNA from fecal samples, performed molecular diagnostic assays, performed chimpanzee microsatellite genotyping analyses, performed phylogenetic analyses, performed all statistical analyses, extracted all remote sensing climate data, extracted all forest cover data, and performed all geospatial mapping, in the laboratories of B.H.H. and M.T.D. W.L., Y.L., and B.H.H. performed molecular diagnostic assays for all other field sites. J.N.N., M.P., S.K., A.E.P., E.V.L., C.M.S., D.B.M., A.K.P., F.A.S., M.K.G., N.S., C.A., K.Z., K.K., C.A.C, R.C., A.R., M.A.H., and R.W.W. facilitated chimpanzee fecal sample collection from field sites across equatorial Africa. N.D.W. provided chimpanzee dried blood spot samples from the Limbe sanctuary in Cameroon. C.A.C. facilitated the collection of climate data within Kibale National Park. R.W.W., M.T.D., and B.H.H. offered revisions of the manuscript. All authors contributed to the intellectual content of this article.

## Competing interests
The authors declare no competing interests.
