## [Peer Review File · Communications Biology]

Reviewers' comments:

Reviewer #1 (Remarks to the Author):

This is a really interesting and well written analysis of *Laverania* species, *Plasmodium* species (primate clade), and *Hepaticystis* detection in chimpanzees and exploration of the relationship between ambient temperature driving spatiotemporal variation in overall infection prevalence.

Major comments:

1. The inherent poor sensitivity of faecal PCR detection for parasite DNA (even with the intensive protocol described) does complicate interpretation of results - particularly for the initial description of true infection species prevalence and diversity. In the referenced Loy et al 2018 paper only 10-20% of those positive for *Plasmodium* spp in blood were also PCR positive in faeces. However this study was conducted in humans and no *Laverania* species were detected so it is hard to extrapolate accuracy for its use in this chimp population (please correct me if this method has been better established/validated than what I suggest here). I agree with the authors however that this means the true infection prevalence is likely vastly underestimated - would it be worth doing a sensitivity analysis around this to estimate variance in real infection prevalence?

2. Limitations in infection prevalence estimates from faecal detection may have some impact on the reliability of the subsequent modelled infection probability analyses, although using both longitudinal and cross-sectional datasets was a good approach to minimise bias related to this. Given the detection constraints, the relative degree of variance in infection prevalence due to temperature/seasonality compared to other factors affecting parasitaemia such as chimp age (as a surrogate of host immunity) is difficult to know. It was unfortunate age wasn't available to also include as a covariate in the larger cross-sectional analysis as well to show they all remained independent predictors.

3. *Hepaticystis* infection linked to human. This is a really interesting finding. Less host restriction within NHP for *Hepaticystis* is a tantalising indicator that spillover zoonotic transmission to humans is possible, although agree this would need confirmation from an infected human blood rather than faecal sample. Direct faecal contamination is possible (or are chimps eaten by humans as bushmeat source in this area as an alternative mechanism?).

4. Seasonality analysis methods. I agree there is likely a seasonal trend in infection however there may be more robust methods to demonstrate this point compared to testing whether the monthly proportion was greater than the dataset mean. A more standard approach to ascertain if time series data is random or associated with adjacent periods would be an autocorrelation plot. Fig 4A is also difficult to interpret as it doesn't show the variance in proportion at each month. Could consider a box plot figure to show that it is greater than 95%CI or similar?

5. Demonstrating goodness of fit of models. Both full GLMM models evaluating ecological drivers of spatiotemporal variation were appropriately tested and superior to the null model, however how well did the final model fit the data? i.e. what was the adjusted r^2 etc as a minimum. Model fit should be documented in Table 2 and 3. For model 2, it was stated 2,346 samples collected but only data points with all covariate data included in model (if this is less than 2,346 then please state final number or change wording to make this clearer).

6. For the cross-sectional dataset, what was the final infection prevalence % and CIs in this dataset? What was the species diversity/breakdown? What proportion had the intensive PCR replicates conducted and was this site / time specific? Were the ages of the chimps sampled known?

7. Covariates in spatiotemporal models - there are a couple of important co-variables that may also be good to explore in these models. The negative rainfall correlation in model 1 is difficult to explain. Given rainfall data was only for the preceding month, when higher rainfall may indicate flushing of

transient oviposition sites, a 2-3 month cumulative total to encompass any lag in increased breeding of mosquitos affecting transmission may be more relevant. Elevation, particularly for model 2 would be important to include. The number/size of chimps within troops, and/or number of troops known at each site would be an important factor influencing transmission intensity to consider (if possible) in model 2.

8. Extrapolating model of infection probability across chimp host range. Infection probability in chimps alone is a very simple estimate/proxy for potential spill-over risk to humans. Probabilistic distributions of varying chimp and human populations (and even vectors) would markedly strengthen this preliminary model. Other models particularly exploring zoonotic malaria spillover to humans for *P. knowlesi* in Southeast Asia have demonstrated forest fragmentation indices (as a marker of human land use change, increased interaction with humans, adaptive vector and NHP behaviour) rather than intact forest cover are a better indicator of spillover risk.

Reviewer #2 (Remarks to the Author):

This article presents a study that aimed to characterize the epidemiology of chimpanzees malarial parasites, with particular attention made for the temperature importance in the transmission dynamics of these *Plasmodium* species. Although the authors put a large emphasis on their observation that their study would give a baseline indicator of human exposure risk (based mostly on the temperature of transmission) several points need to be clarified, added and discussed, as described below in the Major comments section.

Major comments:

1/ This study presents chimpanzee infections by species of the *Laverania* sub-genus, and for part of the samples (not all) by *Hepaticystis*. Why focusing only on chimpanzee *Laverania* infection? Indeed, chimpanzees are also infected by other *Plasmodium* species of the subgenus *Plasmodium*, as the authors show very well in figure 2. When knowing that *P. vivax* in humans is characterized by a broader geographic distribution, associated to lower temperature optimum (in comparison to *P. falciparum*), it would be interesting to add such information in this study (or at least discuss it since the sample size is not very big for the species *P. vivax*-like).

2/ Why the authors chose to present the results of detection of *Hepaticystis* only for part of the samples screened?

3/ It is well known that chimpanzees are characterized by an omnivore diet and that they regularly eat small monkeys' species, some of which are known to be infected by *Hepaticystis* parasites. In such a situation, how could the authors be so sure that the detection of *Hepaticystis* in chimpanzee's faecal samples in their study is not the result of the detection of parasites of monkey's species that chimpanzees ate? How could we test such a hypothesis and prove that these infections are real chimpanzee's infections? Authors really need to prove this and moderate their statement and conclusion about host specificity.

4/ Line 112, page 3: the authors wrote that '*P. falciparum* in the human population has been traced to an ancient gorilla-to-human transmission event (>10,000 years ago'. This is one possible hypothesis, based on the observation that all *P. falciparum* coalesced together before merging with *P. praefalciparum* and of the fact that their diversity is entirely included inside the known diversity of *P. praefalciparum* (Liu et al. 2010; Sundararaman et al. 2013). Noteworthy, the elements originally used to propose that *P. falciparum* evolved following the transmission of a single parasite also can be questioned for several reasons. For instance, *P. falciparum* is much less diverse and that in phylogenetic trees it forms a single well-supported clade within the radiation of the *P. praefalciparum* sequences are not necessarily explained by a single parasite transfer. A scenario in which *P. falciparum* originated from multiple gorilla-to-human transfers followed by a strong bottleneck (i.e. reduction in the population size) more recently in time also could have led to such phylogenetic pattern).

I will not discuss here in more details the arguments for this part (this is not the purpose of this

paper), but today, the evidence in favour of a single parasite transfer is rather scarce, and the hypothesis of a transfer that involved multiple parasites is favoured. In such a situation where currently, no study really showing anything, the authors need to rewrite accordingly this part of the text concerning this 'single transfer' dated 10,000 years ago). Finally, the authors refer to Otto et. 2016, which is not a paper referring to such dating and suggesting any single jump hypothesis. Please modify accordingly.

5/ Lines 120-128, page 4: The authors need to be careful with their conclusion. Indeed, no study today described any transfer of great apes Plasmodium of the Plasmodium subgenus to humans. Although cross-species transmission of Laverania has been documented in captive environments but only between great apes, or from humans to great apes and not the opposite. This part needs to be more explicative of this important difference, which is very important for the author discussion.

6/ Lines 129-133, page 4: Authors need to add/present the other prevalence values detected in other published studies to have some comparative points between studies and sites, methodologies and countries.

7/ Line 167, page 5: the authors explain that they will consider the ambient temperature as a critical driver of spatiotemporal variation on Laverania prevalence. This is very interesting, however, the precipitations are also the other critical driver of such spatiotemporal variation, since without water there will be no breeding habitat for the aquatic mosquito larvae development. The authors need to explain by offering a hypothesis why to their knowledge rainfall is negatively correlated with infection probability in both cross-sectional and sectional studies (not only the temperature effect) for the transmission of these vector-borne parasites.

8/ Lines 172-173, page 5: the authors only amplified Hepatocystis from a subset of samples: why only form a subset? How many samples the authors then screened and for which purpose? Why not relate such information to the temperature as for Plasmodium transmission above?

9/ Lines 182-185, page 6: the conclusion of the authors needs moderation. Such analysis is very important indeed in the knowledge obtained about the epidemiology of these Plasmodium species, however with such high host species specificity in Laverania parasites it is very difficult to consider such study as a baseline indicator of human exposure risk for zoonotic transmission, also when knowing that the genetically closely species of *P. falciparum* in *P. preafalciparum* circulating in gorillas.

10/ Lines 202-206, page 6: The microsatellite markers characterization is a good strategy, with a success of 97.7% on 44 samples. However, this would lead to a miss assignment of 20 individuals for the total 878 samples analyzed... In the next lines, the authors explained that such a misidentification rate is acceptable for the next analysis of demographic predictors. The authors need to justify such a statement with the addition of according references.

11/ Lines 272-283 : The seasonal pattern is clear when considering all Plasmodium species and Hepatocystis infections in chimpanzees of the Kanyawara. What about the seasonal pattern according to the Plasmodium species considered and the co-infection considered? Is there any difference that could be interpreted?

12/ Lines 328-330: the authors need to moderate again such conclusion.

13/ Lines 400-402: this is difficult to validate that such a map would inform efforts to quantify human exposure to these parasites, when no human infection by the Laverania species have been described and when the closely genetic known species of *P. falciparum* is circulating in gorillas and not chimpanzees. In this way, the authors need to put less weight on the output of such results obtained. Additionally, why did the authors did not consider gorilla's infection in their predictive analysis, that would have added a lot more interesting discussion about the exposure of humans to these parasites?

14/ Lines 422-429, pages 13-14: This is not very clear how did the authors consider the effect of forest cover, have all the samples analyzed been collected in the forest?

15/ Lines 442-443, Lines 451-453, page 14: same remark as in 13/.

16/ Lines 474-479, page 15: Authors should add in the discussion the information about the prevalence of *P. vivax*-like and Hepatocystis in relation to the expected ecological trends. Indeed, such discussion would add a lot of value to the paper, especially when knowing that the only human infection with a great ape Plasmodium species was with *P. vivax*-like.

17/ Lines 508-511, page 16: same remark as in 13/.

18/ For the chimpanzee identification, how this was done for the new 500 samples collected for this

study? The same question for the 1,939 other chimpanzee's samples analyzed. This part needs to be more detailed.

19/ Lines 576-577, page 18: why only a subset of pan-African samples was subjected to intensified PCR protocol? How many samples exactly in this subset were considered/analyzed?

Minor comments:

Abstract section:

1/ Line 85, page 3: Specify the chimpanzees' species concerned.

2/ Line 87, page 3: Add known prevalence described in the literature in chimpanzees.

3/ Lines 103-105, page 3: The authors must qualify their conclusions as they do not consider in their conclusion the Hepatocystis infections and the Plasmodium species of the subgenus Plasmodium known to circulate in Africa among chimpanzees.

Authors also need to add the exact number of samples analyzed, and precise the number already obtained data from the literature and the new samples analyzed and then new data generated.

Introduction section:

4/ Lines 115, page 4: Add the two sub-species of chimpanzees considered.

5/ Line 118, page 4: Specify the Anopheles species considered.

Results section:

6/ Line 190, page 6: add references about these recent studies.

7/ Line 216, Page7: 'Of these 878 samples, 31.1% tested', which corresponds to how many samples? This needs to be added.

8/ Line 227, page 7: Add as a supplementary table the list of the reference sequences used with their GenBank number etc.

9/ Line 275, page 9: the proportion of positive samples in all Plasmodium species included?

Hepatocystis included? The authors need to precise on this point.

Materials and methods section:

10/ Line 516, page 16 : 3,314 faecal samples from' how many individuals, this needs to be written.

P. vivax should be written *P. vivax*-like and *P. malariae* should be written *P. malariae*-like.

Figures:

Figure 4. Panel B, precise the signification of OG, NT and LK.

Figure 5. Panel C. Add in the generated all the sample collection sites of this study.

Reviewer #1 (Remarks to the Author):

This is a really interesting and well written analysis of *Laverania* species, *Plasmodium* species (primate clade), and *Hepatocystis* detection in chimpanzees and exploration of the relationship between ambient temperature driving spatiotemporal variation in overall infection prevalence.

Major comments:

1. The inherent poor sensitivity of faecal PCR detection for parasite DNA (even with the intensive protocol described) does complicate interpretation of results - particularly for the initial description of true infection species prevalence and diversity. In the referenced Loy et al 2018 paper only 10-20% of those positive for *Plasmodium* spp in blood were also PCR positive in faeces. However this study was conducted in humans and no *Laverania* species were detected so it is hard to extrapolate accuracy for its use in this chimp population (please correct me if this method has been better established/validated than what I suggest here). I agree with the authors however that this means the true infection prevalence is likely vastly underestimated - would it be worth doing a sensitivity analysis around this to estimate variance in real infection prevalence?

Response: We agree that this study's focus on fecal PCR detection, rather than blood parasitemia, is an important caveat of this study, as highlighted in the results section of this paper. We also agree that a study that replicates Loy et al. (2018) in chimpanzee hosts would constitute a useful follow-up and would add context to the analyses presented herein. Unfortunately, this analysis is beyond the scope of this study and would not be possible with the samples currently in possession. Wild chimpanzees inhabiting long-term observational field sites are generally subject to non-invasive opportunistic collection protocols, which generally restrict invasive blood sample collection. The analyses of non-invasive samples presented in this study offer unique insight into a difficult to address topic, and we believe that these assertions are properly caveated.

2. Limitations in infection prevalence estimates from faecal detection may have some impact on the reliability of the subsequent modelled infection probability analyses, although using both longitudinal and cross-sectional datasets was a good approach to minimise bias related to this. Given the detection constraints, the relative degree of variance in infection prevalence due to temperature/seasonality compared to other factors affecting parasitaemia such as chimp age (as a surrogate of host immunity) is difficult to know. It was unfortunate age wasn't available to also include as a covariate in the larger cross-sectional analysis as well to show they all remained independent predictors.

Response: Agreed—given the strong effect of age in the longitudinal analysis, it is likely that the unavailability of this variable in the cross-sectional analysis introduced a degree of statistical noise. However, we cannot think of any reason to speculate that there would be any directional bias that would conflate the observed relationship with ambient temperature (e.g., no reason to think that communities living in areas ~25°C would be biased to be younger, and thus more likely to test positive). To the contrary, the fact that the effect of ambient temperature on PCR+ probability was discernible despite this noise testifies to the strength of this relationship.

3. *Hepatocystis* infection linked to human. This is a really interesting finding. Less host restriction within NHP for *Hepatocystis* is a tantalising indicator that spillover zoonotic transmission to humans is possible, although agree this would need confirmation from an infected human blood rather than faecal sample. Direct faecal contamination is possible (or are chimps eaten by humans as bushmeat source in this area as an alternative mechanism?).

Response: Yes, we agree that zoonotic ape-to-human hepatocystis transmission is just one possibility for this finding. Chimpanzee predation of *Hepatocystis*-positive monkeys is certainly an alternative explanation, and follow-up studies are needed to address the latter. We have included this caveat in the Results and Discussion sections of this revised draft. However, it is worth noting that we failed in the initial draft to indicate that one *Hepatocystis* isolate was derived from the blood sample of a sanctuary chimpanzee, indicating that these parasites are, indeed, capable of generating productive infection in chimpanzees and that fecal contamination cannot serve as an alternative explanation in all cases. These caveats have been added to the revised draft.

4. Seasonality analysis methods. I agree there is likely a seasonal trend in infection however there may be more robust methods to demonstrate this point compared to testing whether the monthly proportion was greater than the dataset mean. A more standard approach to ascertain if time series data is random or associated with adjacent periods would be an autocorrelation plot. Fig 4A is also difficult to interpret as it doesn't show the variance in proportion at each month. Could consider a box plot figure to show that it is greater than 95%CI or similar?

Response: Unfortunately, more robust time series analysis was precluded by the relatively spotty nature of sample collection in the longitudinal analysis (see table below, which depicts the monthly number of samples collected during the study period at Kanyawara). In Figure 4A, we simply pooled years and evaluated the proportion that tested

positive during each calendar month due to this heterogeneity in sampling (thus no error bars). We agree that this is a crude approach, but the primary takeaway point in the paper is that seasonality is ultimately driven by temporal heterogeneity in environmental factors (e.g., ambient temperature). Infection probability is driven by variation in these underlying environmental factors (as demonstrated by the GLMMs), and the apparent seasonal signature observed in calendar month is a byproduct. In other words, there is nothing “special” about the calendar month beyond the underlying environmental drivers. In essence, figure 4A is more descriptive.

	Jan	Feb	Mar	Apr	May	Jun	Jul	Aug	Sep	Oct	Nov	Dec
2013						38	62	19				
2014						44	11					
2015	30	26	23	31	32	40	62	39	43	41	48	48
2016	38	44	44	40		17	34	24				

5. Demonstrating goodness of fit of models. Both full GLMM models evaluating ecological drivers of spatiotemporal variation were appropriately tested and superior to the null model, however how well did the final model fit the data? i.e. what was the adjusted r2 etc as a minimum. Model fit should be documented in Table 2 and 3. For model 2, it was stated 2,346 samples collected but only data points with all covariate data included in model (if this is less than 2,346 then please state final number or change wording to make this clearer).

Response: Thank you for this comment. In contrast to a GLM, adjusted R² is not an appropriate goodness-of-fit statistic for a GLMM. In the latter, relative measures (e.g., comparison of AIC in full versus null model) are often used to quantify goodness-of-fit. This value is presented in the text. Alternatively, we could present the log likelihood or deviance if desired. The 2436 number references is, indeed, the sample size for which all covariate data was known (and, thus, was included in the model). We have clarified the language in the text. This breakdown is further explored in Figure S2 (which was accidentally omitted from the original text, but has now been included in the revised manuscript).

6. For the cross-sectional dataset, what was the final infection prevalence % and CIs in this dataset? What was the species diversity/breakdown? What proportion had the intensive PCR replicates conducted and was this site / time specific? Were the ages of the chimps sampled known?

Response: The requested summary statistics (prevalence, intensive PCR percentage, etc.) are included in Figure S1-S2. These supplementary tables were mistakenly omitted from the original submission, but are now included in the revised manuscript. The species breakdown was not analyzed in the cross-sectional dataset, because the intensive PCR was not conducted in a substantial proportion of samples (only presence/absence of Laverania was noted). As highlighted in the response to Q2 above, the age of the chimpanzee subjects were not noted in the cross-sectional analyses. In many circumstances, these cross-sectional samples were collected opportunistically during chimpanzee habitat surveys (i.e., in almost all cases, the particular subject was not directly observed, but the species origin of the sample was confirmed by host mtDNA analysis of the fecal sample).

7. Covariates in spatiotemporal models - there are a couple of important co-variables that may also be good to explore in these models. The negative rainfall correlation in model 1 is difficult to explain. Given rainfall data was only for the proceeding month, when higher rainfall may indicate flushing of transient oviposition sites, a 2-3 month cumulative total to encompass any lag in increased breeding of mosquitos affecting transmission may be more relevant. Elevation, particularly for model 2 would be important to include. The number/size of chimps within troops, and/or number of troops known at each site would be an important factor influencing transmission intensity to consider (if possible) in model 2.

Response: The ecological variables analyzed in this study were chosen based upon a relatively deep body of literature (outlined in the introduction). We chose not to include elevation because it is highly correlated with the environmental variables analyzed in this study. Because there is not a strong rationale for an effect of elevation that is independent of ambient temperature/rainfall, its inclusion in the GLMM would only confound the observed ecological relationships. Perhaps a follow-up study, using an alternative approach (e.g., leveraging machine learning) could analyze the effects of dozens of candidate explanatory variables on infection probability, but we chose a more targeted approach, and the former is beyond the scope of this study.

8. Extrapolating model of infection probability across chimp host range. Infection probability in chimps alone is a very simple estimate/proxy for potential spill-over risk to humans. Probabilistic distributions of varying chimp and human populations (and even vectors) would markedly strengthen this preliminary model. Other models particularly exploring

zoonotic malaria spillover to humans for *P. knowlesi* in Southeast Asia have demonstrated forest fragmentation indices (as a marker of human land use change, increased interaction with humans, adaptive vector and NHP behaviour) rather than intact forest cover are a better indicator of spillover risk.

Response: We very much agree with the assertion that there is more to the spillover story than reservoir prevalence, and that human and vector population distributions are critical in predicting exposure. Our “risk map” is not really a true proxy for spillover risk. However, it is also worth noting that this risk map was intended to identify areas a potential high transmission intensity in chimpanzees. Given that previous studies have demonstrated that the Anopheline vectors that transmit these parasites also bite humans, areas associated with high transmission intensity in chimpanzees is likely associated with comparatively higher exposure probability in humans. One can be frequently exposed to a parasite even though molecular barriers limit transmission/spillover. In essence, this risk map is only a first step, and only accounts for the reservoir component. Also, it only articulates spatial heterogeneity in fecal PCR prevalence. To translate this map into a per capita measure of relative risk, more research needs to be done to evaluate spatial heterogeneity in vector density, identify the particular species of Anopheles vectors responsible for transmission across different geographies, and elucidate vector biting preferences. Incorporating human population density would enable one to move beyond relative risk and instead develop a proxy for absolute risk. We have moderated the language in several places, as requested.

Reviewer #2 (Remarks to the Author):

This article presents a study that aimed to characterize the epidemiology of chimpanzee malarial parasites, with particular attention made for the temperature importance in the transmission dynamics of these Plasmodium species. Although the authors put a large emphasis on their observation that their study would give a baseline indicator of human exposure risk (based mostly on the temperature of transmission) several points need to be clarified, added and discussed, as described below in the Major comments section.

Major comments:

1/ This study presents chimpanzee infections by species of the *Laverania* sub-genus, and for part of the samples (not all) by *Hepatocystis*. Why focusing only on chimpanzee *Laverania* infection? Indeed, chimpanzees are also infected by other Plasmodium species of the subgenus Plasmodium, as the authors show very well in figure 2. When knowing that *P. vivax* in humans is characterized by a broader geographic distribution, associated to lower temperature optimum (in comparison to *P. falciparum*), it would be interesting to add such information in this study (or at least discuss it since the sample size is not very big for the species *P. vivax*-like).

Response: This study focused on the *Laverania*, because these species were most prevalent in the fecal samples studied, offering a sample size sufficient for statistical analysis. Infections with *P. vivax/ovale/malariae*-like parasites (as well as *Hepatocystis*) were rare, and thus, the capacity to evaluate these relationships in the other parasites would require a much larger sample size. However, we certainly agree with the reviewer that these relationships would very much be worth investigating in a follow-up study.

2/ Why the authors chose to present the results of detection of *Hepatocystis* only for part of the samples screened?

Response: *Hepatocystis* cytB sequences presented in the longitudinal study (Figure 1 orange; Figure 3 red) were isolated by screening the Kanyawara fecal samples using the SGA methodology outlined in the study. *Hepatocystis* cytB sequences presented in the cross-sectional (Pan-African) study (Figure 3 purple, blue, orange, green) were amplified fortuitously during previous studies (predominantly Liu et al. 2010, 2014, 2016; see Table 1). These *Hepatocystis* sequences have not yet been published.

3/ It is well known that chimpanzees are characterized by an omnivore diet and that they regularly eat small monkeys' species, some of which are known to be infected by *Hepatocystis* parasites. In such a situation, how could the authors be so sure that the detection of *Hepatocystis* in chimpanzee's faecal samples in their study is not the result of the detection of parasites of monkey's species that chimpanzees ate? How could we test such a hypothesis and prove that these infections are real chimpanzee's infections? Authors really need to prove this and moderate their statement and conclusion about host specificity.

Response: We agree that monkey-to-ape *hepatocystis* transmission is just one possibility for this finding. Chimpanzee predation of *Hepatocystis*-positive monkeys is certainly an alternative explanation, and follow-up studies are needed to address the latter. We have included this caveat in the Results and Discussion sections of this revised draft. However, it is worth noting that we failed in the initial draft to indicate that one *Hepatocystis* isolate was derived from the blood sample of a sanctuary chimpanzee, indicating that these parasites are, indeed, capable of generating

productive infection in chimpanzees and that consumption of *Hepaticystis* infected monkeys cannot serve as an alternative explanation in all cases. These caveats have been added to the revised draft.

4/ Line 112, page 3: the authors wrote that 'P. falciparum in the human population has been traced to an ancient gorilla-to-human transmission event (>10,000 years ago'. This is one possible hypothesis, based on the observation that all P. falciparum coalesced together before merging with P. praefalciparum and of the fact that their diversity is entirely included inside the known diversity of P. praefalciparum (Liu et al. 2010; Sundararaman et al. 2013). Noteworthy, the elements originally used to propose that P. falciparum evolved following the transmission of a single parasite also can be questioned for several reasons. For instance, P. falciparum is much less diverse and that in phylogenetic trees it forms a single well-supported clade within the radiation of the P. praefalciparum sequences are not necessarily explained by a single parasite transfer. A scenario in which P. falciparum originated from multiple gorilla-to-human transfers followed by a strong bottleneck (i.e. reduction in the population size) more recently in time also could have led to such phylogenetic pattern). I will not discuss here in more details the arguments for this part (this is not the purpose of this paper), but today, the evidence in favour of a single parasite transfer is rather scarce, and the hypothesis of a transfer that involved multiple parasites is favoured. In such a situation where currently, no study really showing anything, the authors need to rewrite accordingly this part of the text concerning this 'single transfer' dated 10,000 years ago). Finally, the authors refer to Otto et. 2016, which is not a paper referring to such dating and suggesting any single jump hypothesis. Please modify accordingly.

Response: The timing of the emergence of *P. falciparum* in humans and the number of gorilla parasite transmissions are a subject of debate, but as the reviewer states, not the focus of this manuscript. We have thus modified the text, to say "the emergence of *P. falciparum* in the human population has been traced to a gorilla parasite and added the appropriate citations.

5/ Lines 120-128, page 4: The authors need to be careful with their conclusion. Indeed, no study today described any transfer of great apes Plasmodium of the Plasmodium subgenus to humans. Although cross-species transmission of Laverania has been documented in captive environments but only between great apes, or from humans to great apes and not the opposite. This part needs to be more explicative of this important difference, which is very important for the author discussion.

Response: Please see Responses 9 and 12.

6/ Lines 129-133, page 4: Authors need to add/present the other prevalence values detected in other published studies to have some comparative points between studies and sites, methodologies and countries.

Response: Liu et al. (2010) was referenced, because it was the most comprehensive survey of great ape malaria infection, and included surveys of 46 chimpanzee field sites across equatorial Africa. However, additional references have been added to the revised draft.

7/ Line 167, page 5: the authors explain that they will consider the ambient temperature as a critical driver of spatiotemporal variation on Laverania prevalence. This is very interesting, however, the precipitations are also the other critical driver of such spatiotemporal variation, since without water there will be no breeding habitat for the aquatic mosquito larvae development. The authors need to explain by offering a hypothesis why to their knowledge rainfall is negatively correlated with infection probability in both cross-sectional and sectional studies (not only the temperature effect) for the transmission of these vector-borne parasites.

Response: In this section, we had initially attempted to highlight the complex relationship between precipitation and Anopheline population dynamics with the statement: "Precipitation is predicted to be positively correlated with transmission—an effect partially attributable to variation in the availability of breeding habitat for aquatic mosquito larvae (e.g., transitory oviposition sites) (Craig et al., 1999; Parham & Michael, 2010)—though high levels of precipitation may flush breeding sites, resulting in larval mortality (Paaijmans et al., 2007)." However, we have modified this statement to clarify as follows: The effect of precipitation on malaria transmission is more complex. Rainfall is necessary to support breeding habitat for aquatic mosquito larvae (e.g., transitory oviposition sites) (Craig et al., 1999; Parham & Michael, 2010), whereas high levels of precipitation may flush breeding sites, resulting in larval mortality (Paaijmans et al., 2007).

8/ Lines 172-173, page 5: the authors only amplified Hepaticystis from a subset of samples: why only form a subset? How many samples the authors then screened and for which purpose? Why not relate such information to the temperature as for Plasmodium transmission above?

Response: This question is addressed in responses 1 and 2, above.

9/ Lines 182-185, page 6: the conclusion of the authors needs moderation. Such analysis is very important indeed in the knowledge obtained about the epidemiology of these Plasmodium species, however with such high host species specificity in Laverania parasites it is very difficult to consider such study as a baseline indicator of human exposure risk for zoonotic transmission, also when knowing that the genetically closely species of P. falciparum in P. praefalciparum circulating in gorillas.

Response: We have moderated the language, as requested. Specifically, we eliminated the phrase “for zoonotic transmission”. We believe that the phrase “hypothetical baseline” (included in the original text) effectively moderates this assertion. Human exposure does not connote ape-to-human transmission. Accordingly, discussion of molecular barriers to transmission or the host-specificity of the Laverania are not particularly relevant to this assertion. Exposure is dependent upon the prevalence in the reservoir and biting preferences on the Anopheline vector.

10/ Lines 202-206, page 6: The microsatellite markers characterization is a good strategy, with a success of 97.7% on 44 samples. However, this would lead to a miss assignment of 20 individuals for the total 878 samples analyzed... In the next lines, the authors explained that such a misidentification rate is acceptable for the next analysis of demographic predictors. The authors need to justify such a statement with the addition of according references.

Response: Misclassification would introduce statistical noise into the analysis, but there is no reason to think that it would lead to a directional bias, and by extension a spurious result (e.g., no reason to think that older positive chimps would be disproportionately mistaken for younger chimps, resulting in a spurious relationship with age). To the contrary, if anything, the capacity to discern these relationships despite occasional misclassification testifies to the strength of the relationships discerned.

11/ Lines 272-283 : The seasonal pattern is clear when considering all Plasmodium species and Hepatocystis infections in chimpanzees of the Kanyawara. What about the seasonal pattern according to the Plasmodium species considered and the co-infection considered? Is there any difference that could be interpreted?

Response: We have chosen not to investigate epidemiological comparisons amongst parasite species due to limitations of the molecular screening technique used in this study. The single genome amplification protocol outlined in Liu et al. 2010 (referenced in this article) ensures that samples are sufficiently diluted such that mtDNA from only a single parasite is amplified and sequenced per replicate. However, if mitochondrial copy number varies amongst parasites within fecal samples (e.g., if P. gaboni tends to generate higher parasitemia than P. reichenowi, and if this higher parasitemia translates into higher P. gaboni mtDNA copy number in feces), infections caused by the lower copy number species could be masked by that of the higher copy species. In other words, we can be confident that samples that test positive for species A were, indeed, positive for species A. However, we cannot necessarily be confident that samples that tested positive for species A were not also positive for species B (albeit at a relatively lower copy number, resulting in its dilution beyond the limit of detection by the SGA protocol). Accordingly, we have pooled Laverania infections together for analysis.

12/ Lines 328-330: the authors need to moderate again such conclusion.

Response: We have moderated the language, as requested. Specifically, we added the phrase “...and that the apparent host-specificity of the Laverania may be more attributable to cellular—rather than ecological—barriers to infection.” However, it is worth noting that this phrase may be unnecessary, because the host-specificity referenced by the reviewer relates to transmission, whereas lines 328-330 references exposure. The high host-specificity of the Laverania indicates that either human exposure to these parasites is low or the cellular barriers to infection are high. Our analysis demonstrates that prevalence is generally high in the reservoir, and Makanga et al. 2016 indicates that Anopheline vectors exhibit promiscuous biting preferences. Thus, we assert that transmission is limited by these cellular barriers. We claim that exposure is likely high, not that humans are at risk of zoonotic infection.

13/ Lines 400-402: this is difficult to validate that such a map would inform efforts to quantify human exposure to these parasites, when no human infection by the Laverania species have been described and when the closely genetic known species of P. falciparum is circulating in gorillas and not chimpanzees. In this way, the authors need to put less weight on the output of such results obtained. Additionally, why did the authors did not consider gorilla's infection in their predictive analysis, that would have added a lot more interesting discussion about the exposure of humans to these parasites?

Response: As requested, we have clarified the first paragraph of the discussion by modifying the final sentence to “estimation of transmission intensity in chimpanzees could inform efforts to quantify the risk of human exposure to these parasites”. As discussed above, this was intended to identify areas a potential high transmission intensity in

chimpanzees. Given that previous studies have demonstrated that the Anopheline vectors that transmit these parasites also bite humans, areas associated with high transmission intensity in chimpanzees is likely associated with comparatively higher exposure probability in humans. Again, one can be frequently exposed to a parasite even though molecular barrier limit transmission. This analysis could, indeed, “inform efforts to quantify human exposure to these parasites” by identifying these areas of high transmission intensity in the chimpanzee reservoir. Taken further, these may be the areas to target for more comprehensive studies ape-to-human transmission. Were previous studies that found no evidence of ape-to-human transmission simply “looking in the wrong places” or are molecular barriers, indeed, too great for the chimpanzee *Laverania* to overcome? This utility of this analysis is that to provides ecological context that helps to differentiate between these possibilities. Finally, we did not focus on gorilla infections, because this study was intended to be specific to chimpanzee for which we were able to conduct both longitudinal and cross-sectional analyses. However, this would make for an interesting follow-up study, albeit beyond the scope of this analysis.

14/ Lines 422-429, pages 13-14: This is not very clear how did the authors consider the effect of forest cover, have all the samples analyzed been collected in the forest?

Response: The forest cover raster (derived from Hansen et al. 2013) is a continuum from 0% (no forest cover) to 100% (full forest canopy cover). Field sites included in the analysis ranged from 16% to 100% (mean: 77%). These values for all sites are listed in supplementary figure S2, which was mistakenly omitted from the original submission.

15/ Lines 442-443, Lines 451-453, page 14: same remark as in 13/.

Response: As discussed above, human exposure does not connote cross-species transmission. As outlined in this paragraph, it is reasonable to assert that the probability of human exposure is greatest where transmission intensity amongst chimpanzees is greatest (given the promiscuous biting preferences of Anopheline vectors of chimp malaria, confirmed in human landing capture experiments outlined in Makanga et al. 2016).

16/ Lines 474-479, page 15: Authors should add in the discussion the information about the prevalence of *P. vivax*-like and *Hepaticystis* in relation to the expected ecological trends. Indeed, such discussion would add a lot of value to the paper, especially when knowing that the only human infection with a great ape *Plasmodium* species was with *P. vivax*-like.

Response: This study focused on the *Laverania*, because these species were most prevalent in the fecal samples studied, offering a sample size sufficient for statistical analysis. Infections with *P. vivax/ovale/malariae*-like parasites (as well as *Hepaticystis*) were rare, and thus, the capacity to evaluate these relationships in this specific parasite would require a much larger sample size. However, we certainly agree with the reviewer that these relationships would very much be worth investigating in a follow-up study. These sites are where one would should conduct surveillance to confirm the findings of previous studies asserting that cross-species *Laverania* transmission does not occur.

17/ Lines 508-511, page 16: same remark as in 13/.

Response: Please see responses to comments 13, 15, and 16.

18/ For the chimpanzee identification, how this was done for the new 500 samples collected for this study? The same question for the 1,939 other chimpanzee's samples analyzed. This part needs to be more detailed.

Response: These 500+1939 samples reference the cross-sectional analysis. As is stated at the end of the “Chimpanzee Samples” section, “Confirmation of host species was obtained... via amplification and sequencing of host mitochondrial DNA (Etienne et al., 2012; Keele et al., 2006, 2009; Li et al., 2012; Neel et al., 2010; Rudicell et al., 2010) (cross-sectional analyses).” This standard methodology is well-documented in these references.

19/ Lines 576-577, page 18: why only a subset of pan-African samples was subjected to intensified PCR protocol? How many samples exactly in this subset were considered/analyzed?

Response: In the longitudinal analysis, all samples were subjected to the intensified PCR protocol. In the cross-sectional (“Pan-African”) analysis, we predominantly analyzed samples from previously published studies (Liu et al. 2010, 2014, 2016; please see Table 1), which were screened using the methodology outlined in those studies. The number of samples subjected to the intensified PCR protocol is outlined in Figure S2 (column 2), which was unintentionally omitted from the initial submission. We apologize for this omission from the original submission.

Minor comments:

Abstract section:

1/ Line 85, page 3: Specify the chimpanzees' species concerned.

Response: We have added the scientific name in the revised draft.

2/ Line 87, page 3: Add known prevalence described in the literature in chimpanzees.

Response: We chosen to omit these values in the abstract because they are highly variable. This is described in greater depth in the introduction.

3/ Lines 103-105, page 3: The authors must qualify their conclusions as they do not consider in their conclusion the Hepatocystis infections and the Plasmodium species of the subgenus Plasmodium known to circulate in Africa among chimpanzees.

Authors also need to add the exact number of samples analyzed, and precise the number already obtained data from the literature and the new samples analyzed and then new data generated.

Introduction section:

Response: Please see response to Major Comment #13.

4/ Lines 115, page 4: Add the two sub-species of chimpanzees considered.

Response: We have chosen to omit sub-species, because previous studies have demonstrated that all four chimpanzee sub-species harbor Laverania.

5/ Line 118, page 4: Specify the Anopheles species considered.

Response: Anopheles species names have been added to the revised draft.

6/ Line 190, page 6: add references about these recent studies.

Response: Relevant references (cited earlier in the draft) have also now been added to this section of the revised draft.

7/ Line 216, Page7: 'Of these 878 samples, 31.1% tested', which corresponds to how many samples? This needs to be added.

Response: We have revised the manuscript to include this value. This value is also included in supplementary figure S1, which was accidentally omitted from the initial submission.

8/ Line 227, page 7: Add as a supplementary table the list of the reference sequences used with their GenBank number etc.

Response: These reference sequences have now been included as a supplementary table, as requested (see Excel file).

9/ Line 275, page 9: the proportion of positive samples in all Plasmodium species included? Hepatocystis included? The authors need to precise on this point.

Materials and methods section:

Response: This information is listed in the final paragraph of the first section of the results section. It is also listed in Figure 2B, which is referenced in previous sections. We have added another reference to Fig. 2B to this section of the revised draft.

10/ Line 516, page 16 : '3,314 faecal samples from' how many individuals, this needs to be written.

Response: The number of individual chimpanzees included in the cross-sectional analysis is unknown, because most of the field sites studies included chimpanzees that were not habituated to human observation (in contrast to the longitudinal analysis at Kanyawara). We have revised the manuscript to replace "vivax" with "vivax-like".

P. vivax should be written P. vivax-like and P. malariae should be written P. malariae-like.

Response: These revisions have been made, as requested.

Figures:

Figure 4. Panel B, precise the signification of OG, NT and LK.

Response: Referenced chimpanzees are now highlighted as red dots in Figure 4B.

Figure 5. Panel C. Add in the generated all the sample collection sites of this study.

Response: Sample collection sites are listed in Figure 1 with additional details provided in Figure S2.

REVIEWERS' COMMENTS:

Reviewer #2 (Remarks to the Author):

The authors present a longitudinal and cross-sectional study to evaluate if mean ambient temperature drives spatiotemporal variation in chimpanzee *Laverania* infection. They mapped spatiotemporal variation in the suitability of chimpanzee habitat for *Laverania* transmission and gave a hypothetical baseline indicator of human exposure risk, which is a new and interesting approach to consider.

In their revised manuscript, the authors answered all my comments and suggestions really well. I had a good time reading this very interesting research paper. Thank you for this good job.

Virginie Rougeron